# Information Bottleneck for a Rayleigh Fading MIMO Channel with an Oblivious Relay

**Hao Xu** [1] , **Tianyu Yang** [1] , **Giuseppe Caire** [1,*] **and Shlomo Shamai (Shitz)** [2]

1. Faculty of Electrical Engineering and Computer Science, Technical University of Berlin, 10587 Berlin, Germany; xuhao@mail.tu-berlin.de (H.X.); tianyu.yang@tu-berlin.de (T.Y.)
2. Viterbi Electrical Engineering Department, Technion–Israel Institute of Technology, Haifa 32000, Israel; sshlomo@ee.technion.ac.il
* Correspondence: caire@tu-berlin.de

**Abstract:** This paper considers the information bottleneck (IB) problem of a Rayleigh fading multiple-input multiple-out (MIMO) channel with an oblivious relay. The relay is constrained to operating without knowledge of the codebooks, i.e., it performs oblivious processing. Moreover, due to the bottleneck constraint, it is impossible for the relay to inform the destination node of the perfect channel state information (CSI) in each channel realization. To evaluate the bottleneck rate, we first provide an upper bound by assuming that the destination node can obtain a perfect CSI at no cost. Then, we provide four achievable schemes, where each scheme satisfies the bottleneck constraint and gives a lower bound to the bottleneck rate. In the first and second schemes, the relay splits the capacity of the relay–destination link into two parts and conveys both the CSI and its observation to the destination node. Due to CSI transmission, the performance of these two schemes is sensitive to the MIMO channel dimension, especially the channel input dimension. To ensure that it still performs well when the channel dimension grows large, in the third and fourth achievable schemes, the relay only transmits compressed observations to the destination node. Numerical results show that, with simple symbol-by-symbol oblivious relay processing and compression, the proposed achievable schemes work well and can demonstrate lower bounds that come quite close to the upper bound on a wide range of relevant system parameters.

**Keywords:** information bottleneck (IB); oblivious relay; Rayleigh fading; source coding; quantization

## 1. Introduction

For a Markov chain $X \to Y \to Z$ and an assigned joint probability distribution $p_{X,Y}$, consider the following information bottleneck (IB) problem:

$$\max_{p_{Z|Y}} \quad I(X;Z) \tag{1a}$$

$$\text{s.t.} \quad I(Y;Z) \leq C, \tag{1b}$$

where $C$ is the bottleneck constraint parameter and the optimization is with respect to the conditional probability distribution $p_{Z|Y}$ of $Z$ given $Y$. Formulation (1) was introduced by Tishby in [1] and has found remarkable applications in supervised and unsupervised learning problems such as classification, clustering, prediction, etc. [2–7]. From a more fundamental information theoretic viewpoint, the IB arises from the classical remote source coding problem [8–10] under logarithmic distortion [11].

An interesting application of the IB problem in communications consists of a source node, an oblivious relay, and a destination node, which is connected to the relay via an error-free link with capacity $C$. The source node sends codewords over a communication channel and an observation is made at the relay. $X$ and $Y$ are, respectively, the channel input from the source node and output at the relay. The relay is oblivious in the sense that it cannot

decode the information message of the source node itself. This feature can be modeled rigorously by assuming that the source and destination nodes make use of a codebook selected at random over a library, while the relay is unaware of such random selection. For example, in a cloud radio access network (C-RAN), each remote radio head (RRH) acts as a relay and is usually constrained to implement only radio functionalities while the baseband functionalities are migrated to the cloud central processor [12]. Considering the relatively simple structure of the RRHs, it is usually prohibitive to let them know the codebooks and random encoding operations, particularly as the network size becomes large. The fact that the relay cannot decode is also supported by secrecy demands, which means that the codebooks known to the source and destination nodes are to be considered absolutely random, as done here.

Due to the oblivious feature, the relaying strategies that require the codebooks to be known at the relay, e.g., decode-and-forward, compute-and-forward, etc. [13–15], cannot be applied. Instead, the relay has to perform oblivious processing, i.e., employ strategies in the form of compress-and-forward [16–19]. In particular, the relay must treat $X$ as a random process with distribution induced by random selection over the codebook library (see [12] and references therein) and has to produce some useful representation $Z$ by simple signal processing and to convey it to the destination node subject to the link constraint $C$. Then, it makes sense to find $Z$ such that $I(X; Z)$ is maximized.

The IB problem for this kind of communication scenario has been studied in [12,20–26]. In [20], the IB method was applied to reduce the fronthaul data rate of a C-RAN network. References [21,22], respectively, considered Gaussian scalar and vector channels with IB constraint and investigated the optimal tradeoff between the compression rate and the relevant information. In [23], the bottleneck rate of a frequency-selective scalar Gaussian primitive diamond relay channel was examined. In [24,25], the rate-distortion region of a vector Gaussian system with multiple relays was characterized under the logarithmic loss distortion measure. Reference [12] further extended the work in [25] to a C-RAN network with multiple transmitters and multiple relays and studied the capacity region of this network. However, all of References [12,20–25] considered block fading channels and assumed that the perfect channel state information (CSI) was known at both the relay and the destination nodes. In [26], the IB problem of a scalar Rayleigh fading channel was studied. Due to the bottleneck constraint, it was impossible to inform the destination node of the perfect CSI in each channel realization. An upper bound and two achievable schemes were provided in [26] to investigate the bottleneck rate.

In this paper, we extend the work in [26] to the multiple-input multiple-out (MIMO) channel with independent and identically distributed (i.i.d.) Rayleigh fading. This model is relevant for the practical setting of the uplink of a wireless multiuser system where $K$ users send coded uplink signals to a base station. The base station is formed by an RRH with $M$ antennas, connected to a cloud central processor via a digital link of rate $C$ (bottleneck link). The RRH is oblivious to the user codebooks and can apply only simple localized signal processing corresponding to the low-level physical layer functions (i.e., it is an oblivious relay). In current implementations, the RRH quantizes both the uplink pilot symbols and the data-bearing symbols received from the users on each "resource block" (This corresponds roughly to a coherence block of the underlying fading channel in the time-frequency domain) and sends the quantization bits to the cloud processor via the digital link. Here, we simplify the problem, and instead of considering a specific pilot-based channel estimation scheme, we assume that the channel matrix is given perfectly to the relay (remote radiohead), i.e., that the CSI is perfect but local at the relay. Then, we consider an upper bound and specific achievability strategies to maximize the mutual information between the user transmitted signals and the message delivered to the cloud processor, where we allow the relay to operate local oblivious processing as an alternative to direct quantization of both the CSI and the received data-bearing signal.

Intuitively, the relay can split the capacity of the relay-destination link into two parts and convey both the CSI and its observation to the destination node. Hence, in the first

and second achievable schemes, the relay transmits the compressed CSI and observation to the destination node. Specifically, in the first scheme, the relay simply compresses the channel matrix as well as its observation and then forwards them to the destination node. Roughly speaking, this is what happens today in "naive" implementation of RRH systems. Therefore, this scheme can be seen as a baseline scheme. However, the capacity allocated for conveying the CSI to the destination in this scheme is proportional to both the channel input dimension and the number of antennas at the relay. To reduce the channel use required for CSI transmission, in the second achievable scheme, the relay first obtains an estimate of the channel input using channel inversion and then transmits the quantized noise levels as well as the compressed noisy signal to the destination node. In contrast to the first scheme, the capacity allocated to CSI transmission in this scheme is only proportional to the channel input dimension.

Due to the explicit CSI transmission through the bottleneck, the performance of the first and second achievable schemes is sensitive to the MIMO channel dimension, especially the channel input dimension. To ensure that it still performs well when the channel dimension grows large, in the third and fourth achievable schemes, the relay does not convey any CSI to the destination node. In the third scheme, the relay first estimates the channel input using channel inversion and then transmits a truncated representation of the estimate to the destination node. In the fourth scheme, the relay first produces the minimum mean-squared error (MMSE) estimate of the channel input and then source-encodes this estimate. Numerical results show that, with simple symbol-by-symbol oblivious relay processing and compression, the lower bounds obtained by the proposed achievable schemes can come close to the upper bound on a wide range of relevant system parameters.

The rest of this paper is organized as follows. In Section 2, a MIMO channel with Rayleigh fading is presented and the IB problem for this system is formulated. Section 3 provides an upper bound to the bottleneck rate. In Section 4, four achievable schemes are proposed, where each scheme satisfies the bottleneck constraint and gives a lower bound to the bottleneck rate. Numerical results are presented in Section 5 before the conclusions in Section 6.

Throughout this paper, we use the following notations. $\mathbb{R}$ and $\mathbb{C}$ denote the real space and the complex space, respectively. Boldface upper (lower) case letters are used to denote matrices (vectors). $\boldsymbol{I}_K$ stands for the $K \times K$ dimensional identity matrix and $\boldsymbol{0}$ denotes the all-zero vector or matrix. Superscript $(\cdot)^H$ denotes the conjugated-transpose operation, $\mathbb{E}[\cdot]$ denotes the expectation operation, and $[\cdot]^+ \triangleq \max(\cdot, 0)$. $\otimes$ and $\odot$, respectively, denote the Kronecker product and the Hadamard product.

## 2. Problem Formulation

We consider a system with a source node, an oblivious relay, and a destination node as shown in Figure 1. For convenience, we call the source–relay channel "Channel 1" and the relay–destination channel "Channel 2". For Channel 1, we consider the following Gaussian MIMO channel with i.i.d. Rayleigh fading:

$$\boldsymbol{y} = \boldsymbol{H}\boldsymbol{x} + \boldsymbol{n}, \tag{2}$$

where $\boldsymbol{x} \in \mathbb{C}^{K \times 1}$ and $\boldsymbol{n} \in \mathbb{C}^{M \times 1}$ are, respectively, zero-mean circularly symmetric complex Gaussian input and noise with covariance matrices $\boldsymbol{I}_K$ and $\sigma^2 \boldsymbol{I}_M$, i.e., $\boldsymbol{x} \sim \mathcal{CN}(\boldsymbol{0}, \boldsymbol{I}_K)$ and $\boldsymbol{n} \sim \mathcal{CN}(\boldsymbol{0}, \sigma^2 \boldsymbol{I}_M)$. $\boldsymbol{H} \in \mathbb{C}^{M \times K}$ is a random matrix independent of both $\boldsymbol{x}$ and $\boldsymbol{n}$, and the elements of $\boldsymbol{H}$ are i.i.d. zero-mean unit-variance complex Gaussian random variables, i.e., $\boldsymbol{H} \sim \mathcal{CN}(\boldsymbol{0}, \boldsymbol{I}_K \otimes \boldsymbol{I}_M)$. Let $\rho = \frac{1}{\sigma^2}$ denote the signal-to-noise ratio (SNR). Let $\boldsymbol{z}$ denote a useful representation of $\boldsymbol{y}$ produced by the relay for the destination node. $\boldsymbol{x} \to (\boldsymbol{y}, \boldsymbol{H}) \to \boldsymbol{z}$ thus forms a Markov chain. We assume that the relay node has a direct observation of the channel matrix $\boldsymbol{H}$ while the destination node does not since we consider a Rayleigh fading

channel and a capacity-constrained relay–destination link. Then, the IB problem can be formulated as follows:

$$\max_{p(z|y,H)} \quad I(x;z) \tag{3a}$$

$$\text{s.t.} \quad I(y,H;z) \leq C, \tag{3b}$$

where $C$ is the bottleneck constraint, i.e., the link capacity of Channel 2. In this paper, we call $I(x;z)$ the bottleneck rate and $I(y,H;z)$ the compression rate. Obviously, for a joint probability distribution $p(x,y,H)$ determined by (2), problem (3) is a slightly augmented version of IB problem (1). In our problem, we aim to find a conditional distribution $p(z|y,H)$ such that bottleneck constraint (3b) is satisfied and the bottleneck rate is maximized, i.e., as much information on $x$ can be extracted from representation $z$.

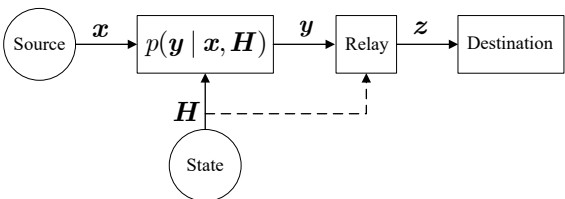

**Figure 1.** Block diagram of the considered information bottleneck (IB) problem.

### 3. Informed Receiver Upper Bound

As stated in [26], an obvious upper bound to problem (3) can be obtained by letting both the relay and the destination node know the channel matrix $H$. We call the bound in this case the informed receiver upper bound. The IB problem in this case takes on the following form:

$$\max_{p(z|y,H)} \quad I(x;z|H) \tag{4a}$$

$$\text{s.t.} \quad I(y;z|H) \leq C. \tag{4b}$$

In Reference [21], the IB problem for a scalar Gaussian channel with block fading has been studied. In the following theorem, we show that, for the considered MIMO channel with Rayleigh fading, (4) can be decomposed into a set of parallel scalar IB problems and the informed receiver upper bound can be obtained based on the result in [21].

**Theorem 1.** *For the considered MIMO channel with Rayleigh fading, the informed receiver upper bound, i.e., the optimal objective function of IB problem (4), is*

$$R^{\text{ub}} = T \int_{\frac{\nu}{\rho}}^{\infty} [\log(1 + \rho\lambda) - \log(1 + \nu)] f_\lambda(\lambda) d\lambda, \tag{5}$$

*where $T = \min\{K, M\}$, $\lambda$ is identically distributed as the unordered positive eigenvalues of $HH^H$; its probability density function (pdf), i.e., $f_\lambda(\lambda)$, is given in (A17); and $\nu$ is chosen such that the following bottleneck constraint is met:*

$$\int_{\frac{\nu}{\rho}}^{\infty} \left( \log \frac{\rho\lambda}{\nu} \right) f_\lambda(\lambda) d\lambda = \frac{C}{T}. \tag{6}$$

**Proof.** See Appendix A.  □

**Lemma 1.** *When $M \to +\infty$ or $\rho \to +\infty$, upper bound $R^{ub}$ tends asymptotically to C. When $C \to +\infty$, $R^{ub}$ approaches the capacity of Channel 1, i.e.,*

$$R^{ub} \to I(x; y, H)$$
$$= T \int_0^\infty \log(1 + \rho\lambda) f_\lambda(\lambda) d\lambda. \tag{7}$$

**Proof.** See Appendix B. □

## 4. Achievable Schemes

In this section, we provide four achievable schemes, where each scheme satisfies the bottleneck constraint and gives a lower bound to the bottleneck rate. In the first and second schemes, the relay transmits both its observation and partial CSI to the destination node. In the third and fourth schemes, to avoid transmitting CSI, the relay first estimates $x$ and then sends a representation of the estimate to the destination node.

*4.1. Non-Decoding Transmission (NDT) Scheme*

Our first achievable scheme assumes that, without decoding $x$, the relay simply source-encodes both $y$ and $H$ and then sends the encoded representations to the destination node. It should be noticed that this scheme is actually reminiscent of the current state-of-the-art in remote antenna head technology, where both the pilot field (corresponding to $H$) and the data field (corresponding to $y$) are quantized and sent to the central processing unit.

Let $h$ denote the vectorization of matrix $H$, and $z_1$ and $z_2$ denote the representations of $h$ and $y$, respectively. From the definition of $H$ in (2), it is known that $h \sim \mathcal{CN}(0, I_{KM})$. Since the elements in $h$ are i.i.d., in the best case, where $I(h; z_1)$ is minimized for a given total distortion, representation $z_1$ introduces the same distortion to each element of $h$. Denote the distortion of each element quantization by $D$. It can then be readily verified by using ([27], Theorem 10.3.3) that the rate distortion function of source $h$ with total squared-error distortion $KMD$ is given by

$$R(D) = \min_{f(z_1|h):\, \mathbb{E}[d(h,z_1)] \leq KMD} I(h; z_1)$$
$$= KM \log \frac{1}{D}, \tag{8}$$

where $0 < D \leq 1$ and $d(h, z_1) = (h - z_1)^H(h - z_1)$ is the squared-error distortion measure. Let $e_1$ denote the error vector of quantizing $h$, i.e., $e_1 = h - z_1$. $z_1$ and $e_1$ are the vectorizations of $Z_1$ and $E_1$. Hence, $H = Z_1 + E_1$. Note that $z_1 \sim \mathcal{CN}(0, (1-D)I_{KM})$, $e_1 \sim \mathcal{CN}(0, DI_{KM})$, and $z_1$ is independent of $e_1$. Hence,

$$\mathbb{E}\left[Z_1 Z_1^H\right] = K(1-D)I_K,$$
$$\mathbb{E}\left[E_1 E_1^H\right] = KDI_K. \tag{9}$$

In ([27], Theorem 10.3.3), the achievability of an information rate for a given distortion, e.g., (8), is proven by considering a backward Gaussian test channel. However, the backward Gaussian test channel does not provide an expression of $z_1$ or $e_1$. Though the specific formulations of $z_1$ and $e_1$ are not necessary for the analysis in this section, since we are providing an achievable scheme, we still give a feasible $z_1$ that satisfies (8) here to make the content more complete. By adding an independent Gaussian noise vector $r \sim \mathcal{CN}(0, \varepsilon I_{KM})$ with $\varepsilon = \frac{D}{1-D}$, to $h$, we get

$$\tilde{h} = h + r. \tag{10}$$

Obviously, $\tilde{h} \sim \mathcal{CN}\left(0, \frac{1}{1-D}I_{KM}\right)$. A representation of $h$ can then be obtained as follows:

$$
\begin{aligned}
z_1 &= \frac{1}{1+\varepsilon}\tilde{h} \\
&= \frac{1}{1+\varepsilon}h + \frac{1}{1+\varepsilon}r \\
&= (1-D)h + (1-D)r,
\end{aligned}
\tag{11}
$$

which is actually the MMSE estimate of $h$ obtained from (10). The error vector is then given by

$$
\begin{aligned}
e_1 &= h - z_1 \\
&= Dh - (1-D)r.
\end{aligned}
\tag{12}
$$

It can be readily verified that $z_1$ provided in (11) satisfies (8), $z_1 \sim \mathcal{CN}(0,(1-D)I_{KM})$, $e_1 \sim \mathcal{CN}(0,DI_{KM})$, and $z_1$ is independent of $e_1$.

To meet the bottleneck constraint, we have to ensure that

$$
I(h,y;z_1,z_2) \leq C.
\tag{13}
$$

Using the chain rule of mutual information,

$$
\begin{aligned}
I(h,y;z_1,z_2) &= I(h,y;z_1) + I(h,y;z_2|z_1) \\
&= I(h;z_1) + I(y;z_1|h) + I(y;z_2|z_1) + I(h;z_2|z_1,y).
\end{aligned}
\tag{14}
$$

Since $z_1$ is a representation of $h$, $y$ and $z_1$ are conditionally independent given $h$. Similarly, since $z_2$ is a representation of $y$, $h$ and $z_2$ are conditionally independent given $y$. Hence,

$$
\begin{aligned}
I(y;z_1|h) &= 0, \\
I(h;z_2|z_1,y) &= 0.
\end{aligned}
\tag{15}
$$

From (8), (14), and (15), it is known that, to guarantee constraint (13), $I(y;z_2|z_1)$, which is the information rate at which the relay quantizes $y$ (given $z_1$), should satisfy

$$
I(y;z_2|z_1) \leq C - R(D).
\tag{16}
$$

Obviously, $C - R(D) > 0$ has to be guaranteed, which yields $D > 2^{-\frac{C}{KM}}$. Hence, in this section, we always assume $2^{-\frac{C}{KM}} < D \leq 1$.

We then evaluate $I(y;z_2|z_1)$. Since $H = Z_1 + E_1$, $y$ in (2) can be rewritten as

$$
\begin{aligned}
y &= Hx + n \\
&= Z_1 x + E_1 x + n.
\end{aligned}
\tag{17}
$$

For a given $Z_1$, the second moment of $y$ is $\mathbb{E}\left[yy^H|Z_1\right] = Z_1 Z_1^H + (KD + \sigma^2)I_M$. Denote the eigendecomposition of $Z_1 Z_1^H$ by $\tilde{U}\Omega\tilde{U}^H$ and

$$
\begin{aligned}
\tilde{y} &= \tilde{U}^H y \\
&= \tilde{U}^H Z_1 x + \tilde{U}^H E_1 x + \tilde{U}^H n.
\end{aligned}
\tag{18}
$$

The second moment of $\tilde{y}$ is $\mathbb{E}\left[\tilde{y}\tilde{y}^H|Z_1\right] = \Omega + (KD + \sigma^2)I_M$. Since $E_1$ is unknown, $\tilde{y}$ is not a Gaussian vector. To evaluate $I(y;z_2|z_1)$, we define a new Gaussian vector

$$
y_g = \tilde{U}^H Z_1 x + n_g,
\tag{19}
$$

where $n_g \sim \mathcal{CN}(\mathbf{0}, (KD + \sigma^2)\mathbf{I}_M)$. For a given $\mathbf{Z}_1$, $\mathbf{y}_g \sim \mathcal{CN}(\mathbf{0}, \mathbf{\Omega} + (KD + \sigma^2)\mathbf{I}_M)$. The channel in (19) can thus be seen as a set of parallel sub-channels. Let $z_g$ denote a representation of $\mathbf{y}_g$, and consider the following IB problem:

$$\max_{p(z_g|y_g)} \quad I(\mathbf{x}; z_g|\mathbf{Z}_1) \tag{20a}$$

$$\text{s.t.} \quad I(\mathbf{y}_g; z_g|\mathbf{Z}_1) \leq C - R(D), \tag{20b}$$

$$2^{-\frac{C}{KM}} < D \leq 1. \tag{20c}$$

Obviously, for a given feasible $D$, problem (20) can be similarly solved as (4) by following the steps in Appendix A. We thus have the following theorem.

**Theorem 2.** *For a given feasible $D$, the optimal objective function of IB problem (20) is*

$$R^{lb1} = T \int_{\frac{v}{\gamma}}^{\infty} [\log(1 + \gamma\lambda) - \log(1 + v)] f_\lambda(\lambda) d\lambda, \tag{21}$$

*where $\gamma = \frac{1-D}{KD+\sigma^2}$; the pdf of $\lambda$, i.e., $f_\lambda(\lambda)$, is given by (A17); and $v$ is chosen such that the following bottleneck constraint is met:*

$$\int_{\frac{v}{\gamma}}^{\infty} \left(\log \frac{\gamma\lambda}{v}\right) f_\lambda(\lambda) d\lambda = \frac{C - R(D)}{T}. \tag{22}$$

**Proof.** See Appendix C. □

Since for a given $\mathbf{Z}_1$, (19) can be seen as a set of parallel scalar Gaussian sub-channels, according to ([21], (16)), the representation of $\mathbf{y}_g$, i.e., $z_g$, can be constructed by adding independent fading and Gaussian noise to each element of $\mathbf{y}_g$. Denote

$$\begin{aligned} z_g &= \mathbf{\Psi}\mathbf{y}_g + \mathbf{n}'_g \\ &= \mathbf{\Psi}\tilde{\mathbf{U}}^H\mathbf{Z}_1\mathbf{x} + \mathbf{\Psi}\mathbf{n}_g + \mathbf{n}'_g, \end{aligned} \tag{23}$$

where $\mathbf{\Psi}$ is a diagonal matrix with nonnegative and real diagonal entries, and $\mathbf{n}'_g \sim \mathcal{CN}(\mathbf{0}, \mathbf{I}_M)$. Note that $\mathbf{y}_g$ in (19) and its representation $z_g$ in (23) are only auxiliary variables. What we are really interested in is the representation of $\mathbf{y}$ and the corresponding bottleneck rate. Hence, we also add fading $\mathbf{\Psi}$ and Gaussian noise $\mathbf{n}'_g$ to $\tilde{\mathbf{y}}$ in (18) and obtain the following representation:

$$\begin{aligned} z_2 &= \mathbf{\Psi}\tilde{\mathbf{y}} + \mathbf{n}'_g \\ &= \mathbf{\Psi}\tilde{\mathbf{U}}^H\mathbf{Z}_1\mathbf{x} + \mathbf{\Psi}\tilde{\mathbf{U}}^H\mathbf{E}_1\mathbf{x} + \mathbf{\Psi}\tilde{\mathbf{U}}^H\mathbf{n} + \mathbf{n}'_g. \end{aligned} \tag{24}$$

In the following lemma, we show that, by transmitting representations $z_1$ and $z_2$ to the destination node, $R^{lb1}$ is an achievable lower bound to the bottleneck rate and the bottleneck constraint is satisfied.

**Lemma 2.** *If the representation of $\mathbf{h}$, i.e., $z_1$ resulting from (8), is forwarded to the destination node for each channel realization, with observations $\mathbf{y}$ and $\mathbf{y}_g$ in (17) and (18) and representations $z_2$ and $z_g$ in (24) and (23), we have*

$$I(\mathbf{y}; z_2|\mathbf{Z}_1) \leq I(\mathbf{y}_g; z_g|\mathbf{Z}_1), \tag{25}$$
$$I(\mathbf{x}; z_2|\mathbf{Z}_1) \geq I(\mathbf{x}; z_g|\mathbf{Z}_1), \tag{26}$$

*where (25) indicates that $I(\mathbf{y}; z_2|\mathbf{Z}_1) \leq C - R(D)$ and (26) gives $I(\mathbf{x}; z_2|\mathbf{Z}_1) \geq R^{lb1}$.*

**Proof.** See Appendix D. □

Lemma 2 shows that, by representing $h$ and $\tilde{y}$ using $z_1$ and $z_2$ in (11) and (24), respectively, lower bound $R^{lb1}$ is achievable and the bottleneck constraint is satisfied.

**Lemma 3.** *When $M \to +\infty$,*

$$R^{lb1} \to T\left[\log(1 + \gamma M) - \log\left(1 + \gamma M 2^{-\frac{C - R(D)}{T}}\right)\right]. \tag{27}$$

*When $\rho \to +\infty$, $R^{lb1}$ tends to a constant, which can be obtained by letting $\gamma = \frac{1-D}{KD}$ and using (21). In addition, when $C \to +\infty$, there exists a small $D$ such that $R^{lb1}$ approaches the capacity of Channel 1, i.e.,*

$$
\begin{aligned}
R^{lb1} &\to I(x; y, H) \\
&= T\int_0^\infty \log(1 + \rho\lambda) f_\lambda(\lambda) d\lambda.
\end{aligned}
\tag{28}
$$

**Proof.** See Appendix E. □

**Remark 1.** *Denote the limit in (27) by $R_0^{lb1} = T\left[\log(1 + \gamma M) - \log\left(1 + \gamma M 2^{-\frac{C - R(D)}{T}}\right)\right]$ for convenience. It can be readily verified that $0 \le R_0^{lb1} \le C$. From (8), it is known that $R(D)$ is also a function of $M$. Moreover, as stated after (16), we always assume $2^{-\frac{C}{KM}} < D \le 1$ in this section such that $C - R(D) > 0$. Hence, when $M \to +\infty$, $D$ approaches 1 and $\gamma$ tends to 0. All this makes it difficult to obtain further concise expression of $R_0^{lb1}$. We investigate the effect of $M$ on $R_0^{lb1}$ in Section 5 by simulation.*

*4.2. Quantized Channel Inversion (QCI) Scheme When $K \le M$*

In our second scheme, the relay first obtains an estimate of the channel input using channel inversion and then transmits the quantized noise levels as well as the compressed noisy signal to the destination node.

In particular, we apply the pseudo inverse matrix of $H$, i.e., $(H^H H)^{-1} H^H$, to $y$ and obtain the zero-forcing estimate of $x$ as follows:

$$
\begin{aligned}
\tilde{x} &= (H^H H)^{-1} H^H y \\
&= x + (H^H H)^{-1} H^H n \\
&\triangleq x + \tilde{n}.
\end{aligned}
\tag{29}
$$

For a given channel matrix $H$, $\tilde{n} \sim \mathcal{CN}(0, A)$, where $A = \sigma^2(H^H H)^{-1}$. Let $A = A_1 + A_2$, where $A_1$ and $A_2$, respectively, consist of the diagonal and off-diagonal elements of $A$, i.e., $A_1 = A \odot I_K$ and $A_2 = A - A_1$. If $H$ could be perfectly transmitted to the destination node, the bottleneck rate could be obtained by following similar steps in Appendix A. However, since $H$ follows a non-degenerate continuous distribution and the bottleneck constraint is finite, as shown in the previous subsection, this is not possible. To reduce the number of bits per channel use required for informing the destination node of the channel information, we only convey a compressed version of $A_1$ and consider a set of independent scalar Gaussian sub-channels.

Specifically, we force each diagonal entry of $A_1$ to belong to a finite set of quantized levels by adding artificial noise, i.e., by introducing physical degradation. We fix a finite grid of $J$ positive quantization points $\mathcal{B} = \{b_1, \cdots, b_J\}$, where $b_1 \le b_2 \le \cdots \le b_{J-1} < b_J$, $b_J = +\infty$, and define the following ceiling operation:

$$\lceil a \rceil_{\mathcal{B}} = \arg\min_{b \in \mathcal{B}}\{a \le b\}. \tag{30}$$

Then, by adding a Gaussian noise vector $\tilde{n}' \sim \mathcal{CN}(0, \text{ diag}\{\lceil a_1 \rceil_{\mathcal{B}} - a_1, \cdots, \lceil a_K \rceil_{\mathcal{B}} - a_K\})$, which is independent of everything else, to (29), a degraded version of $\tilde{x}$ can be obtained

as follows:

$$\hat{x} = \tilde{x} + \tilde{n}'$$
$$= x + \tilde{n} + \tilde{n}'$$
$$\triangleq x + \hat{n}, \tag{31}$$

where $\hat{n} \sim \mathcal{CN}\left(\mathbf{0}, A_1' + A_2\right)$ for a given $H$ and $A_1' \triangleq \mathrm{diag}\{\lceil a_1 \rceil_{\mathcal{B}}, \cdots, \lceil a_K \rceil_{\mathcal{B}}\}$. Obviously, due to $A_2$, the elements in noise vector $\hat{n}$ are correlated.

To evaluate the bottleneck rate, we consider a new variable

$$\hat{x}_g = x + \hat{n}_g, \tag{32}$$

where $\hat{n}_g \sim \mathcal{CN}\left(\mathbf{0}, A_1'\right)$. Obviously, (32) can be seen as $K$ parallel scalar Gaussian sub-channels with noise power $\lceil a_k \rceil_{\mathcal{B}}$ for each sub-channel. Since each quantized noise level $\lceil a_k \rceil_{\mathcal{B}}$ only has $J$ possible values, it is possible for the relay to inform the destination node of the channel information via the constrained link. Note that, from the definition of $A$ in (29), it is known that $a_k$, $\forall k \in \mathcal{K} \triangleq \{1, \cdots, K\}$ are correlated. The quantized noise levels $\lceil a_k \rceil_{\mathcal{B}}$, $\forall k \in \mathcal{K}$ are thus also correlated. Hence, we can jointly source-encode $\lceil a_k \rceil_{\mathcal{B}}$, $\forall k \in \mathcal{K}$ to further reduce the number of bits used for CSI transmission. For convenience, we define a space $\Xi = \{(j_1, \cdots, j_K) | \forall j_k \in \mathcal{J}, k \in \mathcal{K}\}$, where $\mathcal{J} = \{1, \cdots, J\}$. It is obvious that there are a total of $J^K$ points in this space. Let $\xi = (j_1, \cdots, j_K)$ denote a point in space $\Xi$ and define the following probability mass function (pmf):

$$P_\xi = \mathrm{Pr}\{\lceil a_1 \rceil_{\mathcal{B}} = b_{j_1}, \cdots, \lceil a_K \rceil_{\mathcal{B}} = b_{j_K}\}. \tag{33}$$

The joint entropy of $\lceil a_k \rceil_{\mathcal{B}}$, $\forall k \in \mathcal{K}$, i.e., the number of bits used for jointly source-encoding $\lceil a_k \rceil_{\mathcal{B}}$, $\forall k \in \mathcal{K}$, is thus given by

$$H_{\mathrm{joint}} = \sum_{\xi \in \Xi} -P_\xi \log P_\xi. \tag{34}$$

Then, the IB problem for (32) takes on the following form:

$$\max_{p(\hat{z}_g | \hat{x}_g)} \quad I(x; \hat{z}_g | A_1') \tag{35a}$$

$$\mathrm{s.t.} \quad I(\hat{x}_g; \hat{z}_g | A_1') \leq C - H_{\mathrm{joint}}, \tag{35b}$$

where $\hat{z}_g$ is a representation of $\hat{x}_g$.

Note that, as stated above, there are a total of $J^K$ points in space $\Xi$. The pmf $P_\xi$ thus has $J^K$ possible values, and it becomes difficult to obtain the joint entropy $H_{\mathrm{joint}}$ from (34) (even numerically) when $J$ or $K$ is large. To reduce the computational complexity, we consider the (slightly) suboptimal but far more practical entropy coding of each noise level $\lceil a_k \rceil_{\mathcal{B}}$ separately and obtain the following sum of individual entropies:

$$H_{\mathrm{sum}} = \sum_{k=1}^K H_k, \tag{36}$$

where $H_k$ denotes the entropy of $\lceil a_k \rceil_{\mathcal{B}}$ or the number of bits used for informing the destination node of noise level $\lceil a_k \rceil_{\mathcal{B}}$. In Appendix F, we show that $a_k, \forall k \in \mathcal{K}$ are marginally identically inverse chi squared distributed with $M - K + 1$ degrees of freedom and that their pdf is given in (A44). Hence,

$$H_{\text{sum}} = KH_0$$

$$= -K \sum_{j=1}^{J} P_j \log P_j, \tag{37}$$

where $P_j = \text{Pr}\{\lceil a \rceil_{\mathcal{B}} = b_j\}$ can be obtained from (A45) and $a$ follows the same distribution as $a_k$. Since $P_j$ only has $J$ possible values, the computational complexity of calculating $H_{\text{sum}}$ is proportional to $J$. Using the chain rule of entropy and the fact that conditioning reduces entropy, we know that $H_{\text{joint}} \leq H_{\text{sum}}$. In Section 5, the gap between $H_{\text{joint}}$ and $H_{\text{sum}}$ is investigated by simulation. Replacing $H_{\text{joint}}$ in (35b) with $H_{\text{sum}}$, we get the following: IB problem

$$\max_{p(\hat{z}_g|\hat{x}_g)} \quad I(x; \hat{z}_g | A_1') \tag{38a}$$

$$\text{s.t.} \quad I(\hat{x}_g; \hat{z}_g | A_1') \leq C - KH_0. \tag{38b}$$

The optimal solution of this problem is given in the following theorem.

**Theorem 3.** *If $A_1'$ is conveyed to the destination node for each channel realization, the optimal objective function of IB problem (38) is*

$$R^{lb2} = \sum_{j=1}^{J-1} KP_j\big[\log(1 + \rho_j) - \log(1 + \rho_j 2^{-c_j})\big]. \tag{39}$$

*where $\rho_j = \frac{1}{b_j}$, $c_j = \left[\log \frac{\rho_j}{\nu}\right]^+$, and $\nu$ is chosen such that the following bottleneck constraint is met:*

$$\sum_{j=1}^{J-1} KP_j c_j = C - KH_0. \tag{40}$$

**Proof.** See Appendix F. □

Since (32) can be seen as $K$ parallel scalar Gaussian sub-channels, according to ([21], (16)), the representation of $\hat{x}_g$, i.e., $\hat{z}_g$, can be constructed by adding independent fading and Gaussian noise to each element of $\hat{x}_g$. Denote

$$\hat{z}_g = \Phi\hat{x}_g + \hat{n}_g'$$
$$= \Phi x + \Phi\hat{n}_g + \hat{n}_g', \tag{41}$$

where $\Phi$ is a diagonal matrix with positive and real diagonal entries, and $\hat{n}_g' \sim \mathcal{CN}(\mathbf{0}, I_K)$. Note that, similar to $y_g$ and $z_g$ in the previous subsection, $\hat{x}_g$ in (32) and its representation $\hat{z}_g$ in (41) are also auxiliary variables. What we are really interested in is the representation of $\hat{x}$ and the corresponding bottleneck rate. Hence, we also add fading $\Phi$ and Gaussian noise $\hat{n}_g'$ to $\hat{x}$ in (31) and obtain its representation as follows:

$$z = \Phi\hat{x} + \hat{n}_g'$$
$$= \Phi x + \Phi\hat{n} + \hat{n}_g'. \tag{42}$$

In the following lemma, we show that, by transmitting quantized noise levels $\lceil a_k \rceil_{\mathcal{B}}, \forall k \in \mathcal{K}$ and representation $z$ to the destination node, $R^{lb2}$ is an achievable lower bound to the bottleneck rate and the bottleneck constraint is satisfied.

**Lemma 4.** *If $A_1'$ is forwarded to the destination node for each channel realization, with signal vectors $\hat{x}$ and $\hat{x}_g$ in (31) and (32), and their representations $z$ and $\hat{z}_g$ in (42) and (41), we have*

$$I(\hat{x}; z|A_1') \leq I(\hat{x}_g; \hat{z}_g|A_1'), \tag{43}$$

$$I(x; z|A_1') \geq I(x; \hat{z}_g|A_1'), \tag{44}$$

*where (43) indicates that $I(\hat{x}; z|A_1') \leq C - KH_0$ and (44) gives $I(x; z|A_1') \geq R^{lb1}$.*

**Proof.** See Appendix G. □

**Lemma 5.** *When $M \to +\infty$ or $\rho \to +\infty$, we can always find a sequence of quantization points $\mathcal{B} = \{b_1, \cdots, b_J\}$ such that $R^{lb2} \to C$. When $C \to +\infty$,*

$$R^{lb2} \to K\mathbb{E}\left[\log\left(1 + \frac{1}{a}\right)\right]$$

$$\leq I(x; y, H), \tag{45}$$

*where the expectation can be calculated by using the pdf of a in (A44) and $I(x; y, H)$ is the capacity of Channel 1.*

**Proof.** See Appendix H. □

For the sake of simplicity, we may choose the quantization levels as quantiles such that we obtain the uniform pmf $P_j = \frac{1}{J}$. The lower bound (39) can thus be simplified as

$$R^{lb2} = \sum_{j=1}^{J-1} \frac{K}{J} \left[\log(1 + \rho_j) - \log(1 + \rho_j 2^{-c_j})\right], \tag{46}$$

and the bottleneck constraint (40) becomes

$$\sum_{j=1}^{J-1} \left[\log\frac{\rho_j}{\nu}\right]^+ = \frac{JC}{K} - JB, \tag{47}$$

where $B = \log J$ can be seen as the number of bits required for quantizing each diagonal entry of $A_1$. Since $\rho_1 \geq \cdots \geq \rho_{J-1}$, from the strict convexity of the problem, we know that there must exist a unique integer $1 \leq l \leq J - 1$ such that [28]

$$\sum_{j=1}^{l} \log\frac{\rho_j}{\nu} = \frac{JC}{K} - JB,$$

$$\rho_j \leq \nu, \ \forall \, l + 1 \leq j \leq J - 1. \tag{48}$$

Hence, $\nu$ can be obtained from

$$\log\nu = \sum_{j=1}^{l} \frac{\log\rho_j}{l} - \frac{JC}{lK} + \frac{JB}{l}, \tag{49}$$

and $R^{lb1}$ can be calculated as follows:

$$R^{lb2} = \sum_{j=1}^{l} \frac{K}{J} \left[\log(1 + \rho_j) - \log(1 + \nu)\right]. \tag{50}$$

Then, we only need to test the above condition for $l = 1, 2, 3, \cdots$ until (48) is satisfied. Note that, to ensure $R^{lb2} > 0$, $\frac{JC}{K} - JB$ in (47) has to be positive, i.e., $B < \frac{C}{K}$. Moreover, though choosing the quantization levels as quantiles makes it easier to calculate $R^{lb2}$, the results in Lemma 5 may not hold in this case since the choice of quantization points $\mathcal{B} = \{b_1, \cdots, b_J\}$ is restricted.

*4.3. Truncated Channel Inversion (TCI) Scheme When $K \leq M$*

Both the NDT and QCI schemes proposed in the preceding two subsections require that the relay transmits partial CSI to the destination node. Specifically, in the NDT scheme, channel matrix $H$ is compressed and conveyed to the destination node. Hence, the channel use required for transmitting compressed $H$ is proportional to $K$ and $M$. In contrast, the number of bits required for transmitting quantized noise levels in the QCI scheme is proportional to $K$ and $B$. Due to the bottleneck constraint, the performances of the NDT and QCI schemes are thus sensitive to the MIMO channel dimension, especially $K$. To ensure that it still performs well when the channel dimension is large, in this subsection, the relay first estimates $x$ using channel inversion and then transmits a truncated representation of the estimate to the destination node.

In particular, as in the previous subsection, we first obtain the zero-forcing estimate of $x$ using channel inversion, i.e.,

$$
\begin{aligned}
\tilde{x} &= (H^H H)^{-1} H^H y \\
&= x + (H^H H)^{-1} H^H n.
\end{aligned}
\tag{51}
$$

As given in Appendix A, the unordered eigenvalues of $H^H H$ are $\lambda_k$, $\forall k \in \mathcal{K}$. Let $\lambda_{\min} = \min\{\lambda_k, \forall k \in \mathcal{K}\}$. Note that, though the interfering terms can be nulled out by a zero-forcing equalizer, the noise may be greatly amplified when the channel is noisy. Therefore, we put a threshold $\lambda_{\text{th}}$ on $\lambda_{\min}$ such that zero capacity is allocated for states with $\lambda_{\min} < \lambda_{\text{th}}$.

Specifically, when $\lambda_{\min} < \lambda_{\text{th}}$, the relay does not transmit the observation, while when $\lambda_{\min} \geq \lambda_{\text{th}}$, the relay takes $\tilde{x}$ as the new observation and transmits a compressed version of $\tilde{x}$ to the destination node. The information about whether to transmit the observation is encoded into a $0-1$ sequence and is also sent to the destination node. Then, we need to solve the source coding problem at the relay, i.e., encoding blocks of $\tilde{x}$ when $\lambda_{\min} \geq \lambda_{\text{th}}$. For convenience, we use $\Delta$ to denote event "$\lambda_{\min} \geq \lambda_{\text{th}}$". Here, we choose $p(z|\tilde{x}, \Delta)$ to be a conditional Gaussian distribution:

$$
z = \begin{cases} \tilde{x} + q, & \text{if } \Delta \\ \varnothing, & \text{otherwise} \end{cases},
\tag{52}
$$

where $q \sim \mathcal{CN}(0, D I_K)$ is independent of the other variables. It can be easily found from (52) that $I(x; z | \lambda_{\min} < \lambda_{\text{th}}) = 0$ and $I(\tilde{x}; z | \lambda_{\min} < \lambda_{\text{th}}) = 0$. Hence, we consider the following modified IB problem:

$$
\max_{D} \quad P_{\text{th}} I(x; z | \Delta) \tag{53a}
$$

$$
\text{s.t.} \quad P_{\text{th}} I(\tilde{x}; z | \Delta) \leq C - H_{\text{th}}, \tag{53b}
$$

where $P_{\text{th}} = \Pr\{\Delta\}$ and $H_{\text{th}}$ is a binary entropy function with parameter $P_{\text{th}}$.

Since we assume $K \leq M$ in this subsection, as stated in Appendix A, $H^H H \sim \mathcal{CW}_K(M, I_K)$. Then, according to ([29], Proposition 2.6) and ([29], Proposition 4.7), $P_{\text{th}}$ is given by

$$
P_{\text{th}} = \frac{\det \psi}{\prod_{k=1}^{K}(M-k)! \prod_{k=1}^{K}(K-k)!}, \tag{54}
$$

where

$$
\psi = \begin{bmatrix} \psi_0 & \cdots & \psi_{K-1} \\ \vdots & \ddots & \vdots \\ \psi_{K-1} & \cdots & \psi_{2K-2} \end{bmatrix} = \left[ (\psi_{i+j-2}) \right],
$$

$$
\psi_{i+j-2} = \int_{\lambda_{\text{th}}}^{\infty} \mu^{M-K+i+j-2} e^{-\mu} d\mu. \tag{55}
$$

When $K = M$, using ([30], Theorem 3.2), a more concise expression of $P_{\text{th}}$ can be obtained as follows:

$$P_{\text{th}} = \int_{2\lambda_{\text{th}}}^{\infty} \frac{K}{2} e^{-\mu K/2} d\mu$$
$$= e^{-\lambda_{\text{th}} K}. \tag{56}$$

Note that, in (56), the lower bound of the integral is $2\lambda_{\text{th}}$ rather than $\lambda_{\text{th}}$. This is because, in this paper, the elements of $\boldsymbol{H}$ are assumed to be i.i.d. zero-mean unit-variance complex Gaussian random variables, while in [30], the real and imaginary parts of the elements in $\boldsymbol{H}$ are independent standard normal variables.

Given condition $\Delta$, let $\tilde{x}_g$ denote a zero-mean circularly symmetric complex Gaussian random vector with the same second moment as $\tilde{x}$, i.e., $\tilde{x}_g \sim \mathcal{CN}\left(\boldsymbol{0}, \mathbb{E}\left[\tilde{x}\tilde{x}^H | \Delta\right]\right)$, and $\tilde{z}_g = \tilde{x}_g + q$. $P_{\text{th}} I(\tilde{x}_g; \tilde{z}_g | \Delta)$ is then achievable if $P_{\text{th}} I(\tilde{x}_g; \tilde{z}_g | \Delta) \leq C - H_{\text{th}}$. Hence, let

$$P_{\text{th}} I(\tilde{x}_g; \tilde{z}_g | \Delta) = P_{\text{th}} \log \det\left(\boldsymbol{I}_K + \frac{1}{D} \mathbb{E}\left[\tilde{x}\tilde{x}^H | \Delta\right]\right)$$
$$= C - H_{\text{th}}. \tag{57}$$

To calculate $D$ from (57), we denote the eigendecomposition of $\boldsymbol{H}^H\boldsymbol{H}$ by $\boldsymbol{V}\tilde{\boldsymbol{\Lambda}}\boldsymbol{V}^H$, where $\boldsymbol{V}$ is a unitary matrix in which the columns are the eigenvectors of $\boldsymbol{H}^H\boldsymbol{H}$, $\tilde{\boldsymbol{\Lambda}}$ is a diagonal matrix in which the diagonal elements are unordered eigenvalues $\lambda_k$, $\forall k \in \mathcal{K}$, and $\boldsymbol{V}$ and $\tilde{\boldsymbol{\Lambda}}$ are independent. Then, from (51),

$$\mathbb{E}\left[\tilde{x}\tilde{x}^H | \Delta\right] = \boldsymbol{I}_K + \sigma^2 \mathbb{E}\left[(\boldsymbol{H}^H\boldsymbol{H})^{-1} | \Delta\right],$$
$$= \boldsymbol{I}_K + \sigma^2 \mathbb{E}\left[\boldsymbol{V}\tilde{\boldsymbol{\Lambda}}^{-1}\boldsymbol{V}^H | \Delta\right],$$
$$= \boldsymbol{I}_K + \sigma^2 \mathbb{E}\left[\frac{1}{\lambda} | \Delta\right] \boldsymbol{I}_K. \tag{58}$$

Based on [31], the joint pdf of the unordered eigenvalues $\lambda_k$, $\forall k \in \mathcal{K}$ under condition $\Delta$ is given by

$$f(\lambda_1, \cdots, \lambda_K | \Delta) = \frac{1}{P_{\text{th}} K!} \prod_{i=1}^{K} \frac{e^{-\lambda_i} \lambda_i^{M-K}}{(K-i)!(M-i)!} \prod_{i<j}^{K} (\lambda_i - \lambda_j)^2. \tag{59}$$

The marginal pdf of one of the eigenvalues can thus be obtained by integrating out all the other eigenvalues. Taking $\lambda_1$ for example, we have

$$f_{\lambda_1}(\lambda_1 | \Delta) = \int_{\lambda_{\text{th}}}^{\infty} \cdots \int_{\lambda_{\text{th}}}^{\infty} f(\lambda_1, \cdots, \lambda_K | \Delta) d\lambda_2 \cdots d\lambda_K. \tag{60}$$

Then,

$$\mathbb{E}\left[\frac{1}{\lambda} | \Delta\right] = \mathbb{E}\left[\frac{1}{\lambda_1} | \Delta\right]$$
$$= \int_{\lambda_{\text{th}}}^{\infty} \frac{1}{\lambda_1} f_{\lambda_1}(\lambda_1 | \Delta) d\lambda_1. \tag{61}$$

Combining (57), (58), and (61), $D$ can be calculated as follows:

$$D = \frac{1 + \sigma^2 \mathbb{E}\left[\frac{1}{\lambda} | \Delta\right]}{2^{\frac{C-H_{\text{th}}}{P_{\text{th}} K}} - 1}. \tag{62}$$

**Remark 2.** *Note that we show in Appendix I that, when $K = M$ and $\lambda_{th} = 0$, the integral in (61) diverges. $\mathbb{E}\left[\frac{1}{\lambda} | \Delta\right]$ thus does not exist in this case. Therefore, without special instructions, the*

*results derived in this subsection are for the cases with $K = M$ and $\lambda_{th} > 0$ or with $K < M$ and $\lambda_{th} \geq 0$.*

With (57), rate $P_{\text{th}}I(\tilde{x}_g; \tilde{z}_g|\Delta)$ is achievable. Due to the fact that Gaussian input maximizes the mutual information of a Gaussian additive noise channel, we have $I(\tilde{x}; z|\Delta) \leq I(\tilde{x}_g; \tilde{z}_g|\Delta)$. $P_{\text{th}}I(\tilde{x}; z|\Delta)$ is thus also achievable.

The next step is to evaluate the resulting achievable bottleneck rate, i.e., $I(x; z)$. To this end, we first obtain the following lower bound to $I(x; z|\Delta)$ from the fact that conditioning reduces differential entropy,

$$
\begin{aligned}
I(x; z|\Delta) =& h(z|\Delta) - h(z|x, \Delta) \\
\geq& h(z|H, \Delta) - h(z|x, \Delta).
\end{aligned}
\tag{63}
$$

Then, we evaluate the differential entropies $h(z|H, \Delta)$ and $h(z|x, \Delta)$. From (51) and (52), it is known that $z$ is conditionally Gaussian given $H$ and $\Delta$. Hence,

$$
\begin{aligned}
h(z|H, \Delta) =& \mathbb{E}\left[\log(\pi e)^K \det\left(I_K + \sigma^2(H^H H)^{-1} + DI_K\right)|\Delta\right] \\
=& \mathbb{E}\left[\log(\pi e)^K \det\left(I_K + \sigma^2 \tilde{\Lambda}^{-1} + DI_K\right)|\Delta\right] \\
=& K\mathbb{E}\left[\log(\pi e)\left(1 + D + \frac{\sigma^2}{\lambda}\right)|\Delta\right].
\end{aligned}
\tag{64}
$$

On the other hand, using the fact that Gaussian distribution maximizes the entropy over all distributions with the same variance ([27], Theorem 8.6.5), we have

$$
\begin{aligned}
h(z|x, \Delta) =& h(z - x|\Delta) \\
=& h((H^H H)^{-1}H^H n + q|\Delta) \\
\leq& \log(\pi e)^K \det\left(\sigma^2 \mathbb{E}\left[(H^H H)^{-1}|\Delta\right] + DI_K\right) \\
=& K\log(\pi e)\left(D + \sigma^2 \mathbb{E}\left[\frac{1}{\lambda}|\Delta\right]\right).
\end{aligned}
\tag{65}
$$

Substituting (64) and (65) into (63), we can obtain a lower bound to $I(x; z)$, as shown in the following theorem.

**Theorem 4.** *When $K \leq M$, with truncated channel inversion, a lower bound to $I(x; z)$ can be obtained as follows:*

$$
R^{lb3} = P_{th}K\mathbb{E}\left[\log\left(1 + D + \frac{\sigma^2}{\lambda}\right)|\Delta\right] - P_{th}K\log\left(D + \sigma^2 \mathbb{E}\left[\frac{1}{\lambda}|\Delta\right]\right),
\tag{66}
$$

*where $P_{th}$ and $D$ are, respectively, given in (54) and (62), and the expectations can be calculated by using pdf (60).*

**Lemma 6.** *Using Jensen's inequality on convex function $\log(1 + 1/x)$ and concave function $\log x$, we can get a lower bound to $R^{lb3}$, i.e.,*

$$
\check{R}^{lb3} = P_{th}K\log\left(1 + D + \frac{\sigma^2}{\mathbb{E}[\lambda|\Delta]}\right) - P_{th}K\log\left(D + \sigma^2 \mathbb{E}\left[\frac{1}{\lambda}|\Delta\right]\right),
\tag{67}
$$

*and an upper bound to $R^{lb3}$, i.e.,*

$$
\hat{R}^{lb3} = P_{th}K\log\left(1 + D + \sigma^2 \mathbb{E}\left[\frac{1}{\lambda}|\Delta\right]\right) - P_{th}K\log\left(D + \sigma^2 \mathbb{E}\left[\frac{1}{\lambda}|\Delta\right]\right).
\tag{68}
$$

**Remark 3.** *Obviously, $\check{R}^{lb3}$ is also a lower bound to $I(x; z)$. For $\hat{R}^{lb3}$, it is not an upper bound to $I(x; z)$ since it is derived after lower bound $R^{lb3}$. However, we can assess how good the lower bounds $R^{lb3}$ and $\check{R}^{lb3}$ are by comparing them with $\hat{R}^{lb3}$.*

**Lemma 7.** *When $M \to +\infty$, $R^{lb3}$, $\check{R}^{lb3}$, and $\hat{R}^{lb3}$ all tend asymptotically to C. When $\rho \to +\infty$, $R^{lb3}$, $\check{R}^{lb3}$, and $\hat{R}^{lb3}$ all tend asymptotically to $C - H_{th}$. In addition, when $C \to +\infty$, $R^{lb3}$, $\check{R}^{lb3}$, and $\hat{R}^{lb3}$ all approach constants, which can be respectively obtained by setting $D = 0$ in* (66)–(68).

**Proof.** See Appendix J. $\square$

When $K < M$ and $\lambda_{\text{th}} = 0$, it is obvious that $P_{\text{th}} = 1$, $H_{\text{th}} = 0$, and $\mathbb{E}[\lambda] = M$. Since $H^H H \sim \mathcal{CW}_K(M, I_K)$, $(H^H H)^{-1}$ follows a complex inverse Wishart distribution. Hence, $\mathbb{E}\left[\frac{1}{\lambda}\right] = \frac{1}{M-K}$. Then, from Theorem 4 and Lemma 6, we have the following lemma.

**Lemma 8.** *When $K < M$ and $\lambda_{th} = 0$,*

$$R^{lb3} = K\mathbb{E}\left[\log\left(1 + D + \frac{\sigma^2}{\lambda}\right)\right] - K\log\left(D + \frac{\sigma^2}{M-K}\right), \tag{69}$$

$$\check{R}^{lb3} = K\log\left(1 + D + \frac{\sigma^2}{M}\right) - K\log\left(D + \frac{\sigma^2}{M-K}\right), \tag{70}$$

*and*

$$\hat{R}^{lb3} = K\log\left(1 + D + \frac{\sigma^2}{M-K}\right) - K\log\left(D + \frac{\sigma^2}{M-K}\right), \tag{71}$$

*where*

$$D = \frac{1 + \frac{\sigma^2}{M-K}}{2^{\frac{C}{K}} - 1}. \tag{72}$$

**Remark 4.** *When $K < M$, $\lambda_{th} = 0$, and $\frac{\sigma^2}{M-K}$ is small (e.g., when $\rho$ is large, i.e., $\sigma^2$ is small, or when $M - K$ is large), $\hat{R}^{lb3} - \check{R}^{lb3} \approx 0$. In this case, $\check{R}^{lb3}$ is close to $\hat{R}^{lb3}$ and is thus also close to $R^{lb3}$. Then, we can use $\check{R}^{lb3}$ instead of $R^{lb3}$ to lower bound $I(x; z)$ since it has a more concise expression.*

*4.4. MMSE Estimate at the Relay*

In this subsection, we assume that the relay first produces the MMSE estimate of $x$ given $(y, H)$ and then source-encodes this estimate.

Denote

$$F = \left(HH^H + \sigma^2 I_M\right)^{-1} H. \tag{73}$$

The MMSE estimate of $x$ is thus given by

$$\begin{aligned} \bar{x} &= F^H y \\ &= F^H H x + F^H n. \end{aligned} \tag{74}$$

Then, we consider the following modified IB problem:

$$\max_{p(z|\bar{x})} \quad I(x; z) \tag{75a}$$

$$\text{s.t.} \quad I(\bar{x}; z) \leq C. \tag{75b}$$

Note that, since matrix $HH^H + \sigma^2 I_K$ in (73) is always invertible, the results obtained in this subsection always hold no matter $K \leq M$ or $K > M$.

Analogous to the previous subsection, we define

$$
\begin{aligned}
\boldsymbol{z} &= \bar{\boldsymbol{x}} + \boldsymbol{q}, \\
\bar{\boldsymbol{x}}_g &\sim \mathcal{CN}\left(\boldsymbol{0}, \mathbb{E}\left[\bar{\boldsymbol{x}}\bar{\boldsymbol{x}}^H\right]\right), \\
\bar{\boldsymbol{z}}_g &= \bar{\boldsymbol{x}}_g + \boldsymbol{q},
\end{aligned}
\tag{76}
$$

where $\boldsymbol{q}$ has the same definition as in (52), and

$$
\mathbb{E}\left[\bar{\boldsymbol{x}}\bar{\boldsymbol{x}}^H\right] = \mathbb{E}\left[\boldsymbol{F}^H\boldsymbol{H}\boldsymbol{H}^H\boldsymbol{F} + \sigma^2\boldsymbol{F}^H\boldsymbol{F}\right].
\tag{77}
$$

Let

$$
\begin{aligned}
I(\bar{\boldsymbol{x}}_g; \bar{\boldsymbol{z}}_g) &= \log\det\left(\boldsymbol{I}_K + \frac{\mathbb{E}\left[\bar{\boldsymbol{x}}\bar{\boldsymbol{x}}^H\right]}{D}\right) \\
&= C.
\end{aligned}
\tag{78}
$$

Then, rate $I(\bar{\boldsymbol{x}}_g; \bar{\boldsymbol{z}}_g)$ is achievable and $D$ can be calculated from (78). Since $I(\bar{\boldsymbol{x}}; \boldsymbol{z}) \leq I(\bar{\boldsymbol{x}}_g; \bar{\boldsymbol{z}}_g)$, $I(\bar{\boldsymbol{x}}; \boldsymbol{z})$ is thus also achievable.

In the following, we obtain a lower bound to $I(\boldsymbol{x}; \boldsymbol{z})$ by evaluating $h(\boldsymbol{z}|\boldsymbol{H})$ and $h(\boldsymbol{z}|\boldsymbol{x})$ separately and then by using

$$
\begin{aligned}
I(\boldsymbol{x}; \boldsymbol{z}) &= h(\boldsymbol{z}) - h(\boldsymbol{z}|\boldsymbol{x}) \\
&\geq h(\boldsymbol{z}|\boldsymbol{H}) - h(\boldsymbol{z}|\boldsymbol{x}).
\end{aligned}
\tag{79}
$$

First, since $\boldsymbol{z}$ is conditionally Gaussian given $\boldsymbol{H}$, we have

$$
h(\boldsymbol{z}|\boldsymbol{H}) = \mathbb{E}\left[\log(\pi e)^K \det\left(\boldsymbol{F}^H\boldsymbol{H}\boldsymbol{H}^H\boldsymbol{F} + \sigma^2\boldsymbol{F}^H\boldsymbol{F} + D\boldsymbol{I}_K\right)\right].
\tag{80}
$$

Next, based on the fact that conditioning reduces differential entropy and Gaussian distribution maximizes the entropy over all distributions with the same variance [32], we have

$$
\begin{aligned}
h(\boldsymbol{z}|\boldsymbol{x}) &= h(\boldsymbol{z} - \mathbb{E}(\boldsymbol{z}|\boldsymbol{x})|\boldsymbol{x}) \\
&= h\left(\left(\boldsymbol{F}^H\boldsymbol{H} - \mathbb{E}\left[\boldsymbol{F}^H\boldsymbol{H}\right]\right)\boldsymbol{x} + \boldsymbol{F}^H\boldsymbol{n} + \boldsymbol{q}|\boldsymbol{x}\right) \\
&\leq h\left(\left(\boldsymbol{F}^H\boldsymbol{H} - \mathbb{E}\left[\boldsymbol{F}^H\boldsymbol{H}\right]\right)\boldsymbol{x} + \boldsymbol{F}^H\boldsymbol{n} + \boldsymbol{q}\right) \\
&\leq \log(\pi e)^K \det(\boldsymbol{G}),
\end{aligned}
\tag{81}
$$

where

$$
\begin{aligned}
\boldsymbol{G} &= \mathbb{E}\left[\left(\boldsymbol{F}^H\boldsymbol{H} - \mathbb{E}\left[\boldsymbol{F}^H\boldsymbol{H}\right]\right)\left(\boldsymbol{H}^H\boldsymbol{F} - \mathbb{E}\left[\boldsymbol{H}^H\boldsymbol{F}\right]\right) + \sigma^2\boldsymbol{F}^H\boldsymbol{F}\right] + D\boldsymbol{I}_K \\
&= \mathbb{E}\left[\boldsymbol{F}^H\boldsymbol{H}\boldsymbol{H}^H\boldsymbol{F}\right] - \mathbb{E}\left[\boldsymbol{F}^H\boldsymbol{H}\right]\mathbb{E}\left[\boldsymbol{H}^H\boldsymbol{F}\right] + \sigma^2\mathbb{E}\left[\boldsymbol{F}^H\boldsymbol{F}\right] + D\boldsymbol{I}_K.
\end{aligned}
\tag{82}
$$

Combining (79)–(81), we can get a lower bound to $I(\boldsymbol{x}; \boldsymbol{z})$ as shown in the following theorem.

**Theorem 5.** *With the MMSE estimate at the relay, a lower bound to $I(\boldsymbol{x}; \boldsymbol{z})$ can be obtained as follows:*

$$
\begin{aligned}
R^{lb4} = {} & T\mathbb{E}\left[\log\left(\frac{\lambda}{\lambda + \sigma^2} + D\right)\right] + (K - T)\log D \\
& - K\log\left\{\frac{T}{K}\mathbb{E}\left[\frac{\lambda}{\lambda + \sigma^2}\right] - \frac{T^2}{K^2}\left(\mathbb{E}\left[\frac{\lambda}{\lambda + \sigma^2}\right]\right)^2 + D\right\},
\end{aligned}
\tag{83}
$$

*where*

$$D = \frac{\frac{T}{K}\mathbb{E}\left[\frac{\lambda}{\lambda+\sigma^2}\right]}{2^{\frac{C}{K}}-1},$$ (84)

*and the expectations can be calculated by using the pdf of $\lambda$ in (A17).*

**Proof.** See Appendix K. □

**Lemma 9.** *When $M \to +\infty$ or when $K \leq M$ and $\rho \to +\infty$, lower bound $R^{lb4}$ tends asymptotically to C. When $K \leq M$ and $C \to +\infty$,*

$$R^{lb4} \to K\mathbb{E}\left[\log\left(\frac{\lambda}{\lambda+\sigma^2}\right)\right] - K\log\left\{\mathbb{E}\left[\frac{\lambda}{\lambda+\sigma^2}\right] - \left(\mathbb{E}\left[\frac{\lambda}{\lambda+\sigma^2}\right]\right)^2\right\}.$$ (85)

**Proof.** See Appendix L. □

## 5. Numerical Results

In this section, we evaluate the lower bounds obtained by different achievable schemes proposed in Section 4 and compare them with the upper bound derived in Section 3. Before showing the numerical results, we first give the following lemma, which compares the bottleneck rate of the NDT scheme with those of the other three schemes in the $C \to +\infty$ case.

**Lemma 10.** *When $C \to +\infty$, the NDT scheme outperforms the other three schemes, i.e.,*

$$R^{lb1} \geq \max\left\{R^{lb2}, R^{lb3}, R^{lb4}\right\}.$$ (86)

**Proof.** See Appendix M. □

**Remark 5.** *Besides the proof in Appendix M, we can also explain Lemma 10 from a more intuitive perspective. When $C \to +\infty$, the destination node can obtain perfect $\mathbf{y}$ and $\mathbf{H}$ from the relay by using the NDT scheme. The bottleneck rate is thus determined by the capacity of Channel 1. In the QCI scheme, though the destination node can obtain perfect signal vector and noise power of each channel, the correlation between the elements of the noise vector is neglected since the off-diagonal entries of $\mathbf{A}$ are not considered. The bottleneck rate obtained by the QCI scheme is thus upper bounded by the capacity of Channel 1. As for the TCI or MMSE schemes, the destination node can obtain perfect $\tilde{\mathbf{x}}$ or $\bar{\mathbf{x}}$ from the relay. However, the bottleneck rate in these two cases is not only affected by the capacity of Channel 1 but is also limited by the performance of zero-forcing or MMSE estimation since the estimation inevitably incurs a loss of information. Hence, the NDT scheme has a better performance when $C \to +\infty$.*

In the following, we give the numerical results. Note that, when performing the QCI scheme, we choose the quantization levels as quantiles for the sake of convenience.

Figure 2 depicts $R^{lb1}$ versus distortion $D$ under different configurations of SNR $\rho$. It can be found from this figure that $R^{lb1}$ first increases and then decreases with $D$. It is thus important to find a good $D$ to maximize $R^{lb1}$. Since it is difficult to obtain the explicit expression of (21), it is not easy to strictly analyze the relationship between $R^{lb1}$ and $D$. However, we can intuitively explain Figure 2 as follows. When using the NDT scheme, the relay quantizes both $\mathbf{h}$ and $\mathbf{y}$. Due to the bottleneck constraint $C$, there exists a tradeoff. When $D$ is small, the estimation error of $\mathbf{h}$ is small. The destination node can get more CSI, and $R^{lb1}$ thus increases with $D$. When $D$ grows large, though more capacity in $C$ is allocated for quantizing $\mathbf{y}$, the estimation error of $\mathbf{h}$ is large. Hence, $R^{lb1}$ decreases with $D$. In the following simulation process, when implementing the NDT scheme, we vary $D$, calculate $R^{lb1}$ using (21), and then let $R^{lb1}$ be the maximum value.

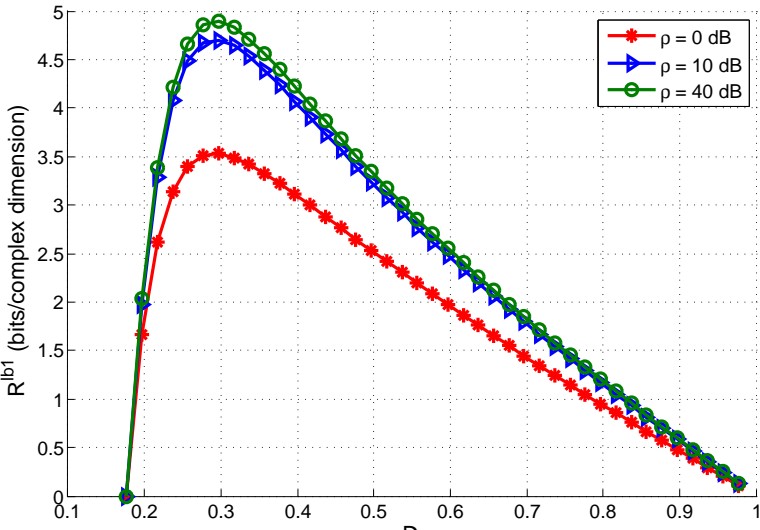

**Figure 2.** Lower bound $R^{lb1}$ versus $D$ with $K = M = 4$ and $C = 40$ bits/complex dimension.

In Figures 3 and 4, we performed Monte Carlo simulations to obtain joint entropy $H_{\text{joint}}$ in (34) and the sum of individual entropies $H_{\text{sum}}$ in (37). Note that, as stated in Section 4.2, the complexities of calculating $H_{\text{joint}}$ and $H_{\text{sum}}$ are, respectively, proportional to $J^K$ and $J$. Hence, when $J$ or $K$ is large, it becomes quite difficult to obtain $H_{\text{joint}}$. For example, when $B = 4$ and $K = 4$, we have $J = 16$ and $J^K = 65,536$, i.e., there are $65,536$ points in space $\Xi$. To obtain a reliable pmf $P_{\zeta}$ for each point, the number of channel realizations has to be much greater than $65,536$.

Figure 3 shows that the gap between $H_{\text{joint}}$ and $H_{\text{sum}}$ is small. In addition, as $M$ increases, $H_{\text{joint}}$ approaches $H_{\text{sum}}$ quickly, indicating that the dependence between $\lceil a_k \rceil_{\mathcal{B}}$, $\forall\, k \in \mathcal{K}$ becomes weak. This can be explained by considering an extreme case where $M \to +\infty$. Based on the definition of $\boldsymbol{H}$ and the strong law of large numbers, we almost surely have $\boldsymbol{H}^H \boldsymbol{H} - M \boldsymbol{I}_K \to \boldsymbol{0}$ when $M \to +\infty$. Hence, $\boldsymbol{A} - \frac{\sigma^2}{M} \boldsymbol{I}_K \to \boldsymbol{0}$. $\lceil a_k \rceil_{\mathcal{B}}$, $\forall\, k \in \mathcal{K}$ are thus almost independent.

When $M = K$ and $K$ increases, Figure 4 shows that there exists an obvious increase in the gap between $H_{\text{joint}}$ and $H_{\text{sum}}$. Hence, when $M = K$ and $K$ increases, the correlation between $\lceil a_k \rceil_{\mathcal{B}}$, $\forall\, k \in \mathcal{K}$ is enhanced. We will thus obtain a gain to $R^{lb2}$ if we use $H_{\text{joint}}$ instead of $H_{\text{sum}}$. However, we would like to point out the following: First, it can be found from Figure 4 that, when $M > K$, this trend becomes less evident. Second, as shown in the following results, when $K \geq 4$, since the QCI scheme uses a lot of capacity in $C$ to quantize $\lceil a_k \rceil_{\mathcal{B}}$, $\forall\, k \in \mathcal{K}$, its performance is not as good as the TCI scheme or MMSE scheme. Third, when $K$ or $B$ is large, it becomes difficult to obtain $H_{\text{joint}}$. Therefore, when implementing the QCI scheme in the following, we obtain $R^{lb2}$ by using $H_{\text{sum}}$, i.e., quantizing $\lceil a_k \rceil_{\mathcal{B}}$, $\forall\, k \in \mathcal{K}$ separately.

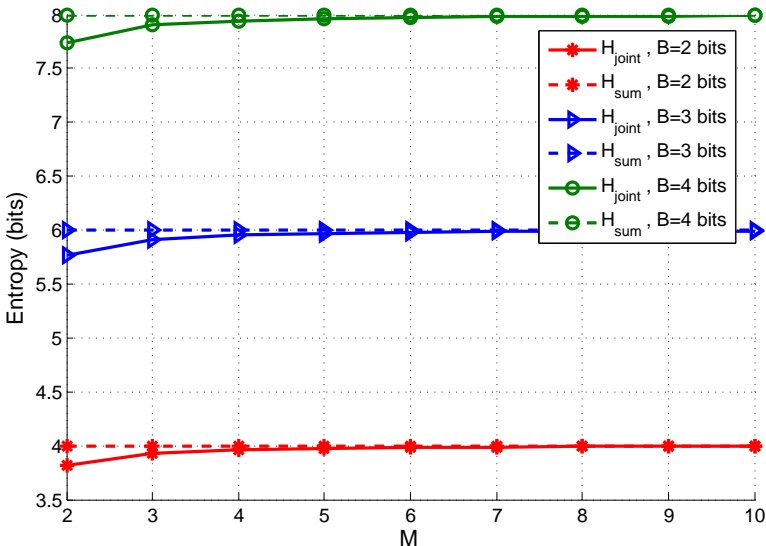

**Figure 3.** $H_{\text{joint}}$ and $H_{\text{sum}}$ versus $M$ with $K = 2$.

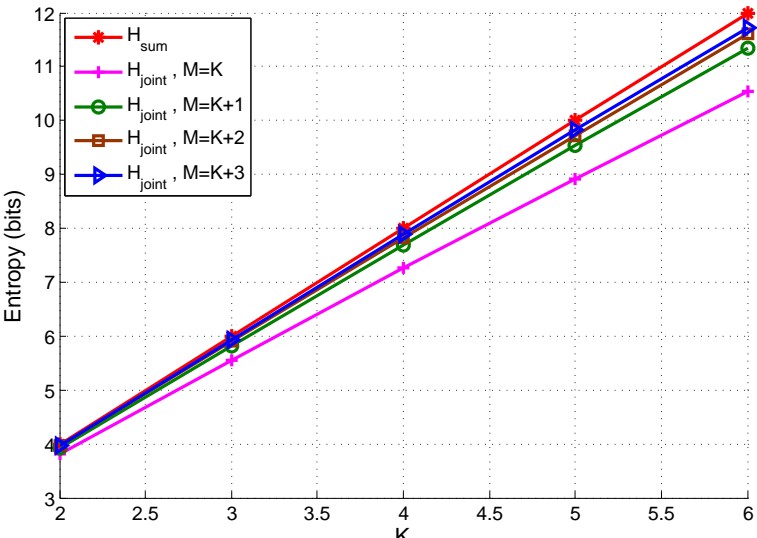

**Figure 4.** $H_{\text{joint}}$ and $H_{\text{sum}}$ versus $K$ with $B = 2$ bits and different values of $M$.

In Figures 5 and 6, we investigate the effect of threshold $\lambda_{\text{th}}$ on $R^{lb3}$ for the cases with $K = M$ and $K < M$, respectively. From these two figures, several observations can be made. First, when $K = M$, and $\rho$ or $K$ is small, $R^{lb3}$ increases greatly and then decreases with $\lambda_{\text{th}}$, indicating that the choice of $\lambda_{\text{th}}$ has a significant impact on $R^{lb3}$. It is thus important to look for a good $\lambda_{\text{th}}$ to maximize $R^{lb3}$ in these cases. Second, when $K = M$, and $K$ as well as $\rho$ are large or when $K < M$, $R^{lb3}$ first remains unchanged and then monotonically decreases with $R^{lb3}$. In these cases, a small $\lambda_{\text{th}}$ is good enough to guarantee a large $R^{lb3}$ and a search for $\lambda_{\text{th}}$ can thus be avoided. For example, when $K < M$, we can set $\lambda_{\text{th}} = 0$, based on which a simpler expression of $R^{lb3}$ is given in (69). For the case with $K = M$, since $\mathbb{E}\left[\frac{1}{\lambda}\right]$ does not exist when $\lambda_{\text{th}} = 0$, we can set $\lambda_{\text{th}}$ to be a fixed small number.

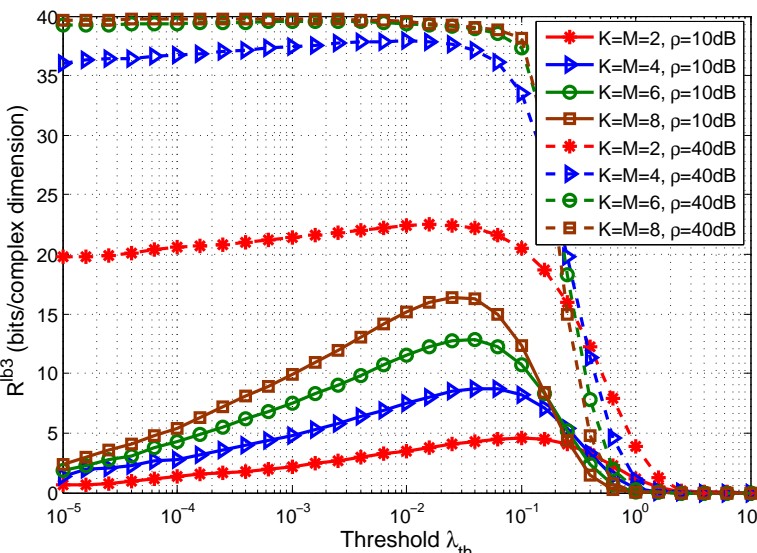

**Figure 5.** Lower bound $R^{lb3}$ versus $\lambda_{th}$ for the $K = M$ case with $C = 40$ bits/complex dimension.

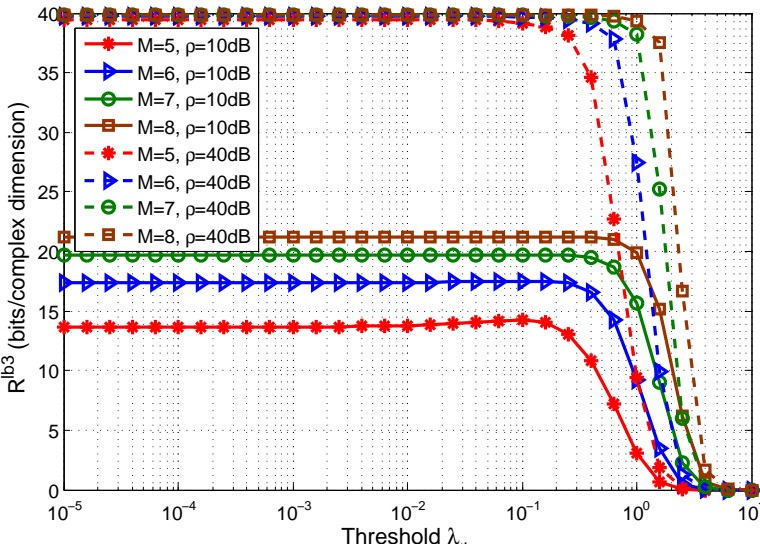

**Figure 6.** Lower bound $R^{lb3}$ versus $\lambda_{th}$ for the $K < M$ case with $K = 4$ and $C = 40$ bits/complex dimension.

In Figures 7 and 8, we compare $R^{lb3}$ with its upper bound $\hat{R}^{lb3}$ and lower bound $\check{R}^{lb3}$. As expected, $R^{lb3}$, $\hat{R}^{lb3}$, and $\check{R}^{lb3}$ all increase with $M$ and $\rho$. When $M$ or $\rho$ is small, there is a small gap between $R^{lb3}$ and $\hat{R}^{lb3}$, and a small gap between $R^{lb3}$ and $\check{R}^{lb3}$. As $M$ and $\rho$ increase, these gaps narrow rapidly and the curves almost coincide, which verifies Remark 2. As a result, when $M - K$ or $\rho$ is large, we can set $\lambda_{th} = 0$ and use $\check{R}^{lb3}$ in (70) to lower bound $I(x; z)$ since it has a more concise expression.

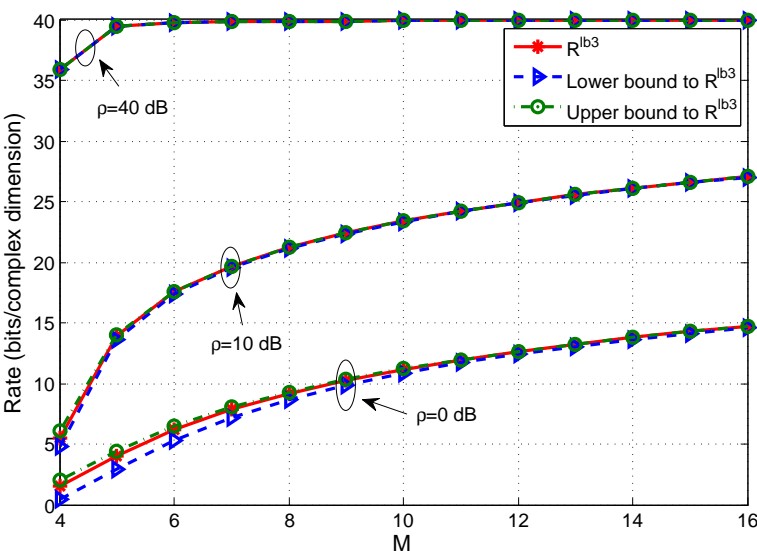

**Figure 7.** $R^{lb3}$, $\check{R}^{lb3}$, and $\hat{R}^{lb3}$ versus $M$ with $K = 4$ and $C = 40$ bits/complex dimension.

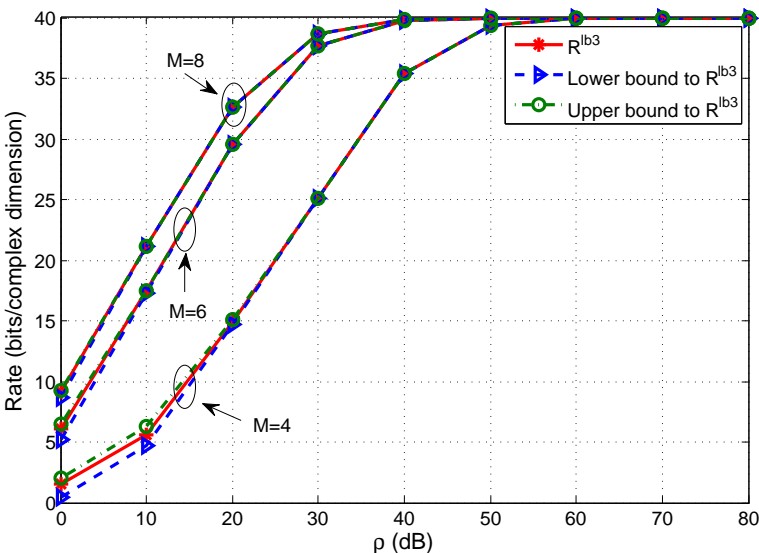

**Figure 8.** $R^{lb3}$, $\check{R}^{lb3}$, and $\hat{R}^{lb3}$ versus $\rho$ with $K = 4$ and $C = 40$ bits/complex dimension.

In Figures 9 and 10, the upper bound $R^{ub}$ and lower bounds obtained by different schemes are depicted versus SNR $\rho$. Several observations can be made from these two figures. First, as expected, all bounds increase with $\rho$. Second, when $K$, $M$, and $\rho$ are small, the NDT scheme outperforms the other achievable schemes. However, as these parameters increase, the performance of the NDT scheme deteriorates rapidly. This is because, when $K$, $M$, and $\rho$ are small, the performance of the considered system is mainly limited by the capacity of Channel 1, and the NDT scheme works well since the destination node can extract more information from the compressed observation of the relay and CSI. However, when $K$ and $M$ increase, the NDT scheme requires too many channel uses for CSI transmission. Third, the QCI scheme can obtain a good performance when $K$ is small. Of course, as stated at the beginning of Section 4.3, the number of bits required for transmitting quantized noise levels in the QCI scheme is proportional to $K$ and $B$. Hence, the performance of the QCI scheme varies significantly when $K$ and $B$ change. Moreover, it is also shown that the performance of the TCI scheme is worse than that of the MMSE scheme in the low SNR regime while getting quite close to that of the MMSE scheme in the high SNR regime. When $\rho$ grows large, the lower bounds obtained by the TCI and MMSE schemes both approach $C$ and are larger than those obtained by the NDT and QCI schemes.

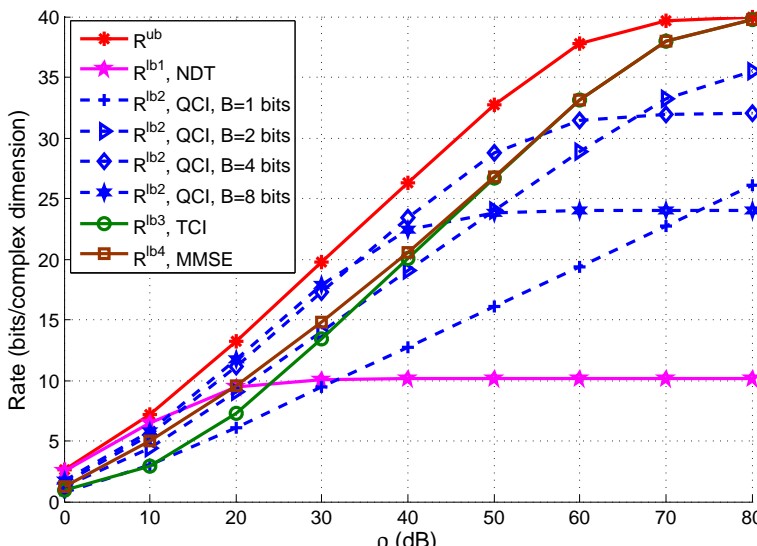

**Figure 9.** Upper and lower bounds to the bottleneck rate versus $\rho$ with $K = M = 2$ and $C = 40$ bits/complex dimension.

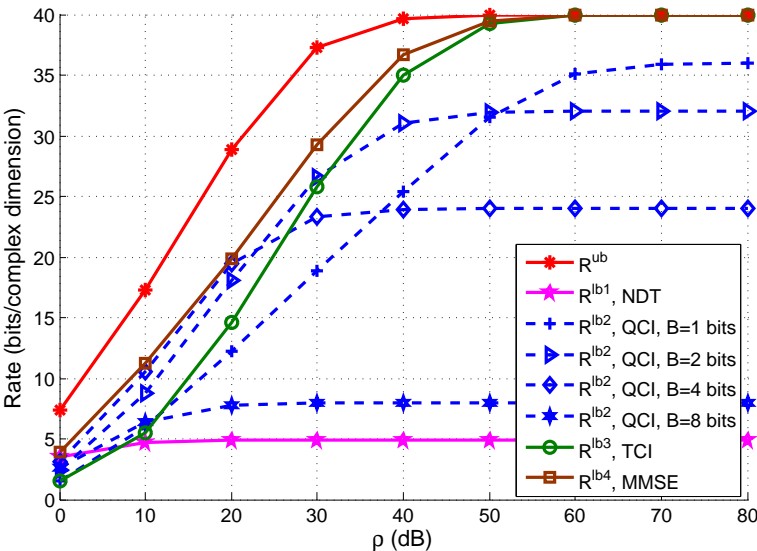

**Figure 10.** Upper and lower bounds to the bottleneck rate versus $\rho$ with $K = M = 4$ and $C = 40$ bits/complex dimension.

In Figures 11 and 12, the effect of the bottleneck constraint $C$ is investigated. From Figure 11, it can be found that, as $C$ increases, all bounds grow and converge to different constants, which can be calculated based on Lemmas 1, 3, 5, 7 and 9. Figure 11 also shows that, thanks to CSI transmission, the NDT and QCI schemes outperform the TCI and MMSE schemes when $C$ is large. By comparing these two figures, it can be found that, in Figure 11, no bound approaches $C$, even for the case with $C = 20$, while in Figure 12, it is possible for $R^{ub}$, $R^{lb3}$, and $R^{lb4}$ to approach $C$. For example, when $K = M = 4$ and $C \leq 30$, $R^{ub}, R^{lb3}, R^{lb4} \to C$. This is because the bottleneck rate is limited by the capacity of Channel 1 and $C$. In Figure 11, since $K$ and $M$ are small, the capacity of Channel 1 is smaller than $C$. Hence, the bounds of course will not approach $C$. In Figure 12, more multi-antenna gains can be obtained due to larger $K$ and $M$. The capacity of Channel 1 is thus larger than $C$ in some cases (e.g., $K = M = 4$ and $C \leq 30$). Hence, $R^{ub}$, $R^{lb3}$, and $R^{lb4}$ may approach $C$ in these cases. Note that, as shown in Figure 11, since $B < \frac{C}{K}$ is not satisfied, $R^{lb4} = 0$ when $C \leq 30$.

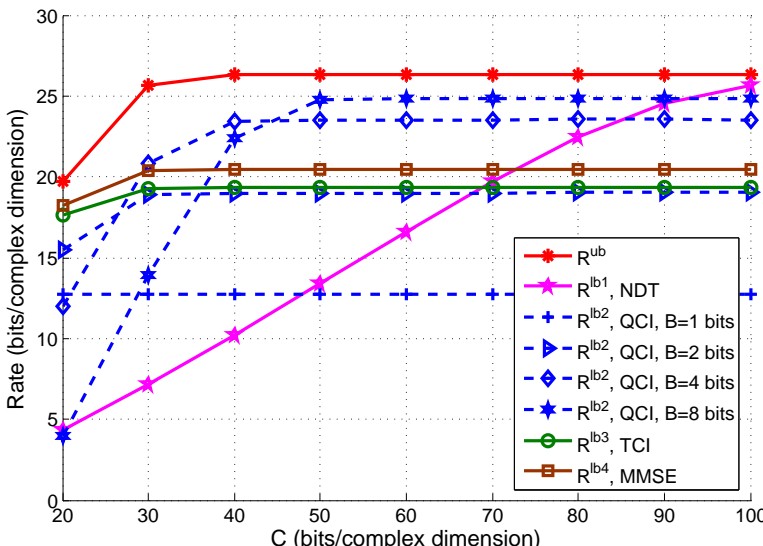

**Figure 11.** Upper and lower bounds to the bottleneck rate versus $C$ with $K = M = 2$ and $\rho = 40$ dB.

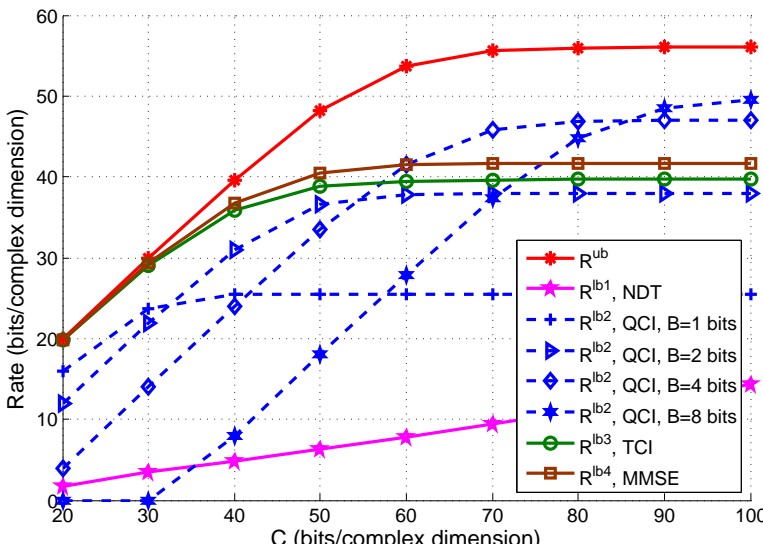

**Figure 12.** Upper and lower bounds to the bottleneck rate versus $C$ with $K = M = 4$ and $\rho = 40$ dB.

In Figures 13 and 14, the effect of $M$ is investigated for different configurations of $\rho$. These two figures show that $R^{ub}$, $R^{lb2}$, $R^{lb3}$, and $R^{lb4}$ all increase monotonically with $M$ and that, as $M$ grows, $R^{lb3}$ as well as $R^{lb4}$ become very close to $R^{ub}$. For $R^{lb1}$, except the $M = 3$ case in Figure 13, $R^{lb1}$ monotonically decreases with $M$ since the relay has to transmit more channel information to the destination node.

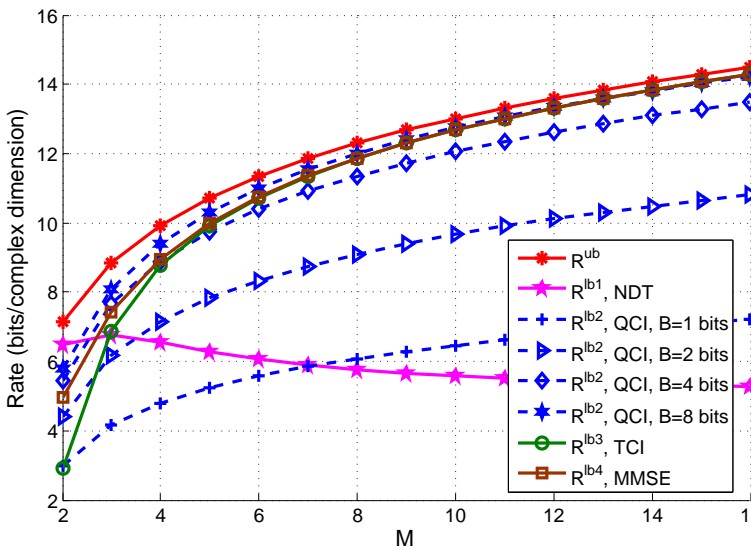

**Figure 13.** Upper and lower bounds to the bottleneck rate versus $M$ with $K = 2$, $\rho = 10$ dB, and $C = 40$ bits/complex dimension.

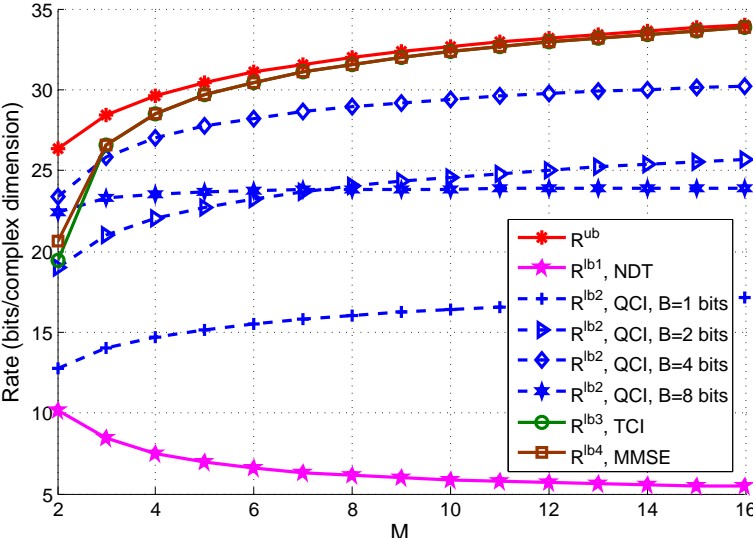

**Figure 14.** Upper and lower bounds to the bottleneck rate versus $M$ with $K = 2$, $\rho = 40$ dB, and $C = 40$ bits/complex dimension.

In Figures 15 and 16, we set $K = M$ and depict the upper and lower bounds versus $K$ or $M$. In Figure 15, we fix $C$ to 50, while in Figure 16, we set $C = 8K$, which makes sense since the bottleneck constraint should scale with the number of degrees of freedom of the input signal $x$. Since we choose the quantization levels as quantiles when performing the QCI scheme, as stated at the end of Section 4.2, $B < \frac{C}{K}$ should be satisfied. Hence, in Figures 15 and 16, we only consider $B = 1, 2, 4$ bits when performing the QCI scheme. When $K = M$ and they grow simultaneously, the capacity of Channel 1 increases due to the muti-antenna gains. Hence, for a fixed $C$, Figure 15 shows that all bounds increase first. When $K$ or $M$ grows large, $R^{lb3}$ and $R^{lb4}$ approach the bottleneck constraint $C$ while $R^{lb2}$ decreases for all values of $B$. This is because the number of bits per channel use required for informing the destination node of $A'_1$ in the QCI scheme is proportional to $K$ while CSI transmission is unnecessary for the TCI and MMSE schemes. For the NDT scheme, since the number of bits required for quantizing $H$ is proportional to both $K$ and $M$, there is only an increase when $K$ grows from 1 to 2. After that, $R^{lb1}$ decreases monotonically and has the worst performance. In contrast, when $C = 8K$, the bottleneck rate of the system is mainly

limited by $C$. Hence, Figure 16 shows that all bounds except $R^{lb1}$ increase almost linearly with $K$ and that $R^{ub}$, $R^{lb3}$, and $R^{lb4}$ are quite close to $C$.

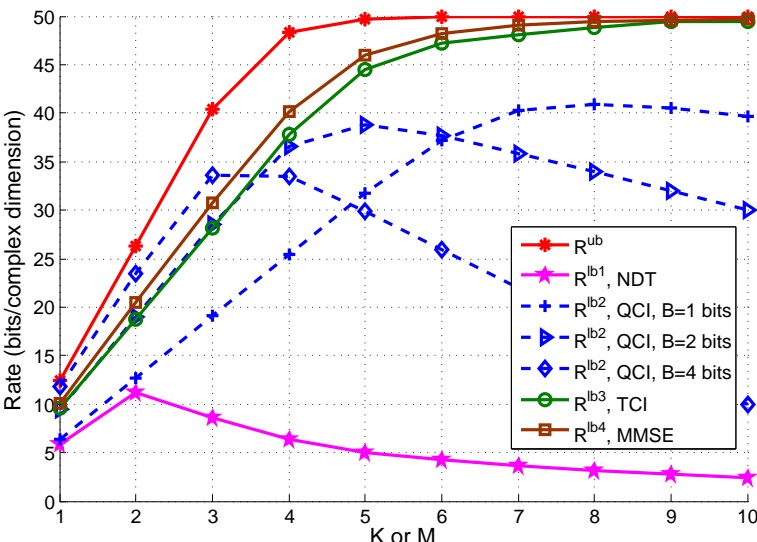

**Figure 15.** Upper and lower bounds to the bottleneck rate versus $K$ or $M$ with $K = M$, $\rho = 40$ dB, and $C = 50$ bits/complex dimension.

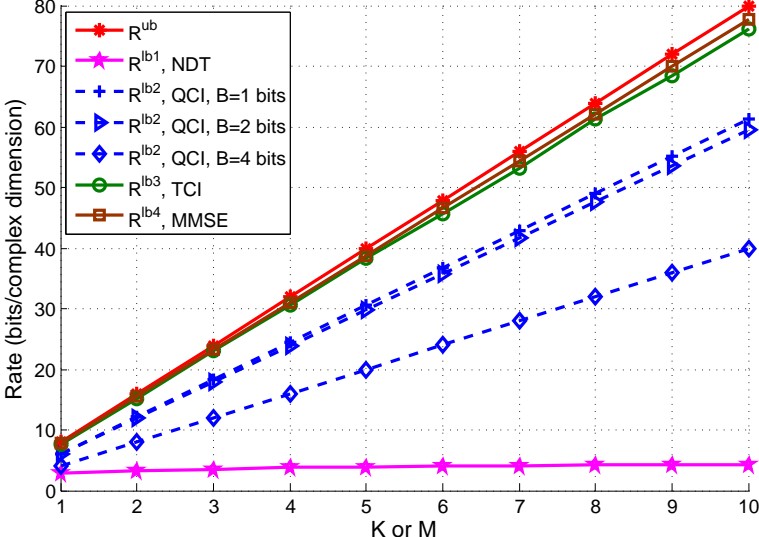

**Figure 16.** Upper and lower bounds to the bottleneck rate versus $K$ or $M$ with $K = M$, $\rho = 40$ dB, and $C = 8K$ bits/complex dimension.

## 6. Conclusions

This work extends the IB problem of the scalar case in [26] to the case of MIMO Rayleigh fading channels. Due to the information bottleneck constraint, the destination node cannot obtain a perfect CSI from the relay. Hence, we provide an upper bound to the bottleneck rate by assuming that the destination node can obtain a perfect CSI at no cost. Moreover, we also provide four achievable schemes, where each scheme satisfies the bottleneck constraint and gives a lower bound to the bottleneck rate. Our results show that, with simple symbol-by-symbol relay processing and compression, we can obtain a bottleneck rate close to the upper bound on a wide range of relevant system parameters. Note that tightening the upper bound is a challenge for future studies. In addition, although we have focused on a MIMO channel with one relay, we plan to extend the problem to considering the case of multiple parallel relays, which is particularly

relevant to the centralized processing of multiple remote antennas, as in the so-called C-RAN architectures.

**Author Contributions:** Conceptualization, H.X., G.C., and S.S.; methodology, H.X., T.Y., G.C., and S.S.; writing—original draft preparation, H.X.; writing—review and editing, H.X., T.Y., G.C., and S.S.; funding acquisition, H.X., G.C., and S.S. All authors have read and agreed to the published version of the manuscript.

**Funding:** This work was supported by the Alexander von Humboldt Foundation and the European Union's Horizon 2020 Research and Innovation Programme with grant agreement No. 694630.

**Institutional Review Board Statement:** Not applicable.

**Informed Consent Statement:** Not applicable.

**Data Availability Statement:** Data sharing not applicable.

**Conflicts of Interest:** The authors declare no conflict of interest.

**Appendix A. Proof of Theorem 1**

Before proving Theorem 1, we first consider the following scalar Gaussian channel:

$$y = sx + n, \tag{A1}$$

where $x \sim \mathcal{CN}(0,1)$, $n \sim \mathcal{CN}(0,\sigma^2)$, and $s \in \mathbb{C}$ is the deterministic channel gain. With bottleneck constraint $C$, the IB problem for (A1) has been studied in [21] and the optimal bottleneck rate is given by

$$R_0 = \log\left(1 + \rho|s|^2\right) - \log\left(1 + \rho|s|^2 2^{-C}\right). \tag{A2}$$

In the following, we show that (4) can be decomposed into a set of parallel scalar IB problems, and (A2) can then be applied to obtain upper bound $R^{\mathrm{ub}}$ in Theorem 1.

According to the definition of conditional entropy, problem (4) can be rewritten as

$$\max_{p(z|y,H)} \quad \int I(x;z|H=\mathbb{H})p_H(\mathbb{H})d\mathbb{H} \tag{A3a}$$

$$\text{s.t.} \quad \int I(y;z|H=\mathbb{H})p_H(\mathbb{H})d\mathbb{H} \leq C, \tag{A3b}$$

where $\mathbb{H}$ is a realization of $H$. Let $U\Lambda U^H$ denote the eigendecomposition of $HH^H$, where $U$ is a unitary matrix in which the columns are the eigenvectors of $HH^H$, and $\Lambda$ is a diagonal matrix in which the diagonal elements are the eigenvalues of $HH^H$. Since the rank of $HH^H$ is no greater than $T = \min\{K,M\}$, there are at most $T$ positive diagonal entries in $\Lambda$. Denote them by $\lambda_t$, where $t \in \mathcal{T}$ and $\mathcal{T} = \{1, \cdots, T\}$. Let

$$\hat{y} = U^H y$$
$$= U^H Hx + U^H n. \tag{A4}$$

Then, for a given channel realization $H = \mathbb{H}$, $\hat{y}$ is conditionally Gaussian, i.e.,

$$\hat{y}|H = \mathbb{H} \sim \mathcal{CN}(0, \Lambda + \sigma^2 I_M). \tag{A5}$$

Since

$$I(x;y|H=\mathbb{H}) = I(x;\hat{y}|H=\mathbb{H}), \tag{A6}$$

we work with $\hat{y}$ instead of $y$ in the following.

Based on (A3) and (A5), it is known that MIMO channel $p(\hat{y}|x,H)$ can be first divided into a set of parallel channels for different realizations of $H$ and that each channel $p(\hat{y}|x,H=\mathbb{H})$ can be further divided into $T$ independent scalar Gaussian channels with

SNRs $\rho\lambda_t, \forall t \in \mathcal{T}$. Accordingly, problem (4) can be decomposed into a set of parallel IB problems. For a scalar Gaussian channel with SNR $\rho\lambda_t$, let $c_t^{\text{ub}}$ denote the allocation of the bottleneck constraint $C$ and $R_t^{\text{ub}}$ denote the corresponding rate. According to (A2), we have

$$R_t^{\text{ub}} = \log(1 + \rho\lambda_t) - \log\left(1 + \rho\lambda_t 2^{-c_t^{\text{ub}}}\right). \tag{A7}$$

Then, the solution of problem (4) can be obtained by solving the following problem:

$$\max_{\{c_t^{\text{ub}}\}} \quad \sum_{t=1}^{T} \mathbb{E}\left[R_t^{\text{ub}}\right] \tag{A8a}$$

$$\text{s.t.} \quad \sum_{t=1}^{T} \mathbb{E}\left[c_t^{\text{ub}}\right] \leq C. \tag{A8b}$$

Assume that $\lambda_t, \forall t \in \mathcal{T}$ are unordered positive eigenvalues of $\boldsymbol{H}\boldsymbol{H}^H$. (Note that, when deriving the upper and lower bounds in this paper, we consider the unordered positive eigenvalues of $\boldsymbol{H}\boldsymbol{H}^H$ or $\boldsymbol{H}^H\boldsymbol{H}$ since it simplifies the analysis. If the ordered positive eigenvalues of $\boldsymbol{H}\boldsymbol{H}^H$ or $\boldsymbol{H}^H\boldsymbol{H}$ are considered, it can be readily proven by following similar steps in ([31], Section 4.2) to arrive at problems equivalent to those in this paper). Then, they are identically distributed. For convenience, define a new variable $\lambda$ that follows the same distribution as $\lambda_t$. The subscript "$t$" in $c_t^{\text{ub}}$ and $R_t^{\text{ub}}$ can thus be omitted. In order to distinguish from $R^{\text{ub}}$ in (5), we use $R_0^{\text{ub}}$ to denote the bottleneck rate corresponding to $c^{\text{ub}}$, i.e.,

$$R_0^{\text{ub}} = \log(1 + \rho\lambda) - \log\left(1 + \rho\lambda 2^{-c^{\text{ub}}}\right). \tag{A9}$$

Then, we have

$$\sum_{t=1}^{T} \mathbb{E}\left[R_t^{\text{ub}}\right] = T\mathbb{E}\left[R_0^{\text{ub}}\right],$$

$$\sum_{t=1}^{T} \mathbb{E}\left[c_t^{\text{ub}}\right] = T\mathbb{E}\left[c^{\text{ub}}\right]. \tag{A10}$$

Problem (A8) is thus equivalent to

$$\max_{c^{\text{ub}}} \quad \mathbb{E}\left[R_0^{\text{ub}}\right] \tag{A11a}$$

$$\text{s.t.} \quad \mathbb{E}\left[c^{\text{ub}}\right] \leq \frac{C}{T}. \tag{A11b}$$

This problem can be solved by the water-filling method. Consider the Lagrangian

$$\mathcal{L} = \mathbb{E}\left[-R_0^{\text{ub}} + \alpha c^{\text{ub}}\right] - \frac{\alpha C}{T}, \tag{A12}$$

where $\alpha$ is the Lagrange multiplier. The Karush-Kuhn-Tucker (KKT) condition for the optimality is

$$\frac{\partial \mathcal{L}}{\partial c^{\text{ub}}} \begin{cases} = 0, & \text{if } c^{\text{ub}} > 0 \\ \leq 0, & \text{if } c^{\text{ub}} = 0 \end{cases}. \tag{A13}$$

Then,

$$c^{\text{ub}} = \begin{cases} \log\frac{\rho\lambda}{\nu}, & \text{if } \lambda > \frac{\nu}{\rho} \\ 0, & \text{if } \lambda \leq \frac{\nu}{\rho} \end{cases}, \tag{A14}$$

where $\nu = \alpha/(1-\alpha)$ and it is chosen such that the following bottleneck constraint is met:

$$\mathbb{E}\left[\log\frac{\rho\lambda}{\nu} \,\Big|\, \lambda > \frac{\nu}{\rho}\right]\Pr\left\{\lambda > \frac{\nu}{\rho}\right\} = \frac{C}{T}. \tag{A15}$$

The informed receiver upper bound is thus given by

$$R^{\text{ub}} = T\mathbb{E}\left[\log(1 + \rho\lambda) - \log(1 + \nu)\,|\lambda > \frac{\nu}{\rho}\right]\Pr\left\{\lambda > \frac{\nu}{\rho}\right\}. \tag{A16}$$

From the definition of $\boldsymbol{H}$ in (2), it is known that, when $K \leq M$ (resp., when $K > M$), $\boldsymbol{H}^H\boldsymbol{H}$ (resp., $\boldsymbol{H}\boldsymbol{H}^H$) is a central complex Wishart matrix with $M$ (resp., $K$) degrees of freedom and covariance matrix $\boldsymbol{I}_K$ (resp., $\boldsymbol{I}_M$), i.e., $\boldsymbol{H}^H\boldsymbol{H} \sim \mathcal{CW}_K(M, \boldsymbol{I}_K)$ (resp., $\boldsymbol{H}\boldsymbol{H}^H \sim \mathcal{CW}_M(K, \boldsymbol{I}_M)$) [33]. Since $\lambda$ can be seen as one of the unordered positive eigenvalues of $\boldsymbol{H}^H\boldsymbol{H}$ or $\boldsymbol{H}\boldsymbol{H}^H$, its pdf is thus given by [31] and ([33], Theorem 2.17):

$$f_\lambda(\lambda) = \frac{1}{T}\sum_{i=0}^{T-1}\frac{i!}{(i+S-T)!}\left[L_i^{S-T}(\lambda)\right]^2\lambda^{S-T}e^{-\lambda}, \tag{A17}$$

where $S = \max\{K, M\}$ and the Laguerre polynomials are

$$L_i^{S-T} = \frac{e^\lambda}{i!\lambda^{S-T}}\frac{d^i}{d\lambda^i}\left(e^{-\lambda}\lambda^{S-T+i}\right). \tag{A18}$$

Substituting (A17) and (A18) into (A16) and (A15), (5) and (6) can be obtained. Theorem 1 is thus proven.

**Appendix B. Proof of Lemma 1**

In order to prove that $R^{\text{ub}}$ approaches $C$ as $M \to +\infty$, we first look at the special case with $K = 1$. In this case, $S = M$ and $T = 1$. From (A18) and (A17), we have $L_0^{S-T} = 1$ and the pdf of $\lambda$

$$f_\lambda(\lambda) = \frac{\lambda^{M-1}e^{-\lambda}}{(M-1)!}, \tag{A19}$$

which shows that $\lambda$ follows Erlang distribution with shape parameter $M$ and rate parameter 1, i.e., $\lambda \sim \text{Erlang}(M, 1)$. The expectation of $\lambda$ is thus $M$. As $M \to +\infty$, $f_\lambda(\lambda)$ becomes a delta function [34]. Hence, for a sufficiently small positive real number $\epsilon$,

$$\lim_{M\to+\infty}\Pr\{|\lambda - M| \leq \epsilon\} \to 1,$$
$$\lim_{M\to+\infty}\Pr\{|\lambda - M| > \epsilon\} \to 0. \tag{A20}$$

Then, when $M \to +\infty$, the bottleneck constraint (6)

$$\int_{\frac{\nu}{\rho}}^\infty\left(\log\frac{\rho\lambda}{\nu}\right)f_\lambda(\lambda)d\lambda = C$$
$$\to \int_{M-\epsilon}^{M+\epsilon}\left(\log\frac{\rho\lambda}{\nu}\right)f_\lambda(\lambda)d\lambda$$
$$\to \log\frac{\rho M}{\nu}, \tag{A21}$$

based on which we get

$$\frac{\nu}{M} \to \rho 2^{-C}. \tag{A22}$$

Using (5), (A20), and (A22), it is known that, when $M \to +\infty$,

$$R^{\mathrm{ub}} = \int_{\frac{\nu}{\rho}}^{\infty} [\log(1+\rho\lambda) - \log(1+\nu)] f_\lambda(\lambda) d\lambda$$

$$\rightarrow \int_{M-\epsilon}^{M+\epsilon} \left( \log \frac{1+\rho\lambda}{1+\nu} \right) f_\lambda(\lambda) d\lambda$$

$$\rightarrow \log \frac{1+\rho M}{1+\nu}$$

$$\rightarrow C. \tag{A23}$$

Next, we consider the general case. For any positive integer $K$, when $M \rightarrow +\infty$, based on the definition of $H$ and the strong law of large numbers, we almost surely have $H^H H - M I_K \rightarrow 0$. Since $HH^H$ and $H^H H$ have the same positive eigenvalues, $\lambda - M \rightarrow 0$ almost surely. (A20) thus also holds for this general case. Then,

$$\int_{\frac{\nu}{\rho}}^{\infty} \left( \log \frac{\rho\lambda}{\nu} \right) f_\lambda(\lambda) d\lambda = \frac{C}{T}$$

$$\rightarrow \int_{M-\epsilon}^{M+\epsilon} \left( \log \frac{\rho\lambda}{\nu} \right) f_\lambda(\lambda) d\lambda$$

$$\rightarrow \log \frac{\rho M}{\nu}, \tag{A24}$$

based on which we get

$$\frac{\nu}{M} \rightarrow \rho 2^{-C/T}. \tag{A25}$$

Hence, when $M \rightarrow +\infty$,

$$R^{\mathrm{ub}} \rightarrow T \int_{\frac{\nu}{\rho}}^{\infty} [\log(1+\rho\lambda) - \log(1+\nu)] f_\lambda(\lambda) d\lambda$$

$$\rightarrow T \int_{M-\epsilon}^{M+\epsilon} \left( \log \frac{1+\rho\lambda}{1+\nu} \right) f_\lambda(\lambda) d\lambda$$

$$\rightarrow T \log \frac{1+\rho M}{1+\nu}$$

$$\rightarrow C. \tag{A26}$$

Now we prove that $R^{\mathrm{ub}}$ approaches $C$ as $\rho \rightarrow +\infty$. From (6), it can be seen that $\int_{\frac{\nu}{\rho}}^{\infty} \left( \log \frac{\rho\lambda}{\nu} \right) f_\lambda(\lambda) d\lambda$ reduces with $\nu$. Therefore, when $\rho \rightarrow +\infty$, to ensure that constraint (6) holds, $\nu$ becomes large. Then, we have

$$R^{\mathrm{ub}} = T \int_{\frac{\nu}{\rho}}^{\infty} [\log(1+\rho\lambda) - \log(1+\nu)] f_\lambda(\lambda) d\lambda$$

$$\rightarrow T \int_{\frac{\nu}{\rho}}^{\infty} [\log(\rho\lambda) - \log\nu] f_\lambda(\lambda) d\lambda$$

$$= C. \tag{A27}$$

In addition, when $C \rightarrow +\infty$, it can be found from (6) that $\nu \rightarrow 0$. Using (5), we can get (7), which is the capacity of Channel 1. This completes the proof.

**Appendix C. Proof of Theorem 2**

For a given $Z_1$, $y_g \sim \mathcal{CN}(0, \Omega + (KD + \sigma^2) I_M)$. Let $\omega$ denote the unordered positive eigenvalue of $Z_1 Z_1^H$. Since the elements in $Z_1$ and $H$, respectively, follow i.i.d.,

$\mathcal{CN}(0, 1 - D)$ and $\mathcal{CN}(0, 1)$, and $\lambda$ is the unordered positive eigenvalue of $\boldsymbol{HH}^H$ as defined in Appendix A, $\omega$ is thus identically distributed as $(1 - D)\lambda$. Then, the pdf of $\omega$ is

$$f_\omega(\omega) = \frac{1}{1 - D} f_\lambda\left(\frac{\omega}{1 - D}\right), \tag{A28}$$

where $f_\lambda$ is the pdf of $\lambda$ and is given in (A17).

For a given feasible $D$, problem (20) can be similarly solved as (4) by following the steps in Appendix A and the optimal solution is

$$R^{lb1} = T \int_{\nu(KD+\sigma^2)}^{\infty} \left[\log\left(1 + \frac{\omega}{KD + \sigma^2}\right) - \log(1 + \nu)\right] f_\omega(\omega) d\omega, \tag{A29}$$

where $\nu$ is chosen such that the following bottleneck constraint is met:

$$\int_{\nu(KD+\sigma^2)}^{\infty} \left[\log\frac{\omega}{\nu(KD + \sigma^2)}\right] f_\omega(\omega) d\omega = \frac{C - R(D)}{T}. \tag{A30}$$

Using (A28), (A29) can be reformulated as

$$\begin{aligned}
R^{lb1} &= T \int_{\nu(KD+\sigma^2)}^{\infty} \left[\log\left(1 + \frac{\omega}{KD + \sigma^2}\right) - \log(1 + \nu)\right] f_\omega(\omega) d\omega \\
&= T \int_{\nu(KD+\sigma^2)}^{\infty} \left[\log\left(1 + \frac{\omega}{KD + \sigma^2}\right) - \log(1 + \nu)\right] \frac{1}{1 - D} f_\lambda\left(\frac{\omega}{1 - D}\right) d\omega \\
&\stackrel{\lambda = \frac{\omega}{1-D}}{=\!=\!=\!=\!=} T \int_{\frac{\nu}{\gamma}}^{\infty} [\log(1 + \gamma\lambda) - \log(1 + \nu)] f_\lambda(\lambda) d\lambda,
\end{aligned} \tag{A31}$$

where $\gamma = \frac{1-D}{KD+\sigma^2}$. Analogously, bottleneck constraint (A30) can be transformed to

$$\int_{\frac{\nu}{\gamma}}^{\infty} \left(\log\frac{\gamma\lambda}{\nu}\right) f_\lambda(\lambda) d\lambda = \frac{C - R(D)}{T}. \tag{A32}$$

Theorem 2 is thus proven.

**Appendix D. Proof of Lemma 2**

We first prove inequation (25).

$$\begin{aligned}
I(\boldsymbol{y}; z_2|\boldsymbol{Z}_1) &= I(\tilde{\boldsymbol{y}}; z_2|\boldsymbol{Z}_1) \\
&= h(z_2|\boldsymbol{Z}_1) - h(z_2|\boldsymbol{Z}_1, \tilde{\boldsymbol{y}}) \\
&\stackrel{(a)}{\leq} \mathbb{E}\left[\log\det\left(\boldsymbol{\Omega\Psi}^2 + (KD + \sigma^2)\boldsymbol{\Psi}^2 + \boldsymbol{I}_M\right)\right] \\
&= I(\boldsymbol{y}_g; z_g|\boldsymbol{Z}_1),
\end{aligned} \tag{A33}$$

where $(a)$ holds since Gaussian distribution maximizes the entropy over all distributions with the same variance. Then, we prove inequation (26). Since for a Gaussian input, Gaussian noise minimizes the mutual information ([27], (9.178)), we have

$$I(\boldsymbol{x}; z_2|\boldsymbol{Z}_1) \geq I(\boldsymbol{x}; z_g|\boldsymbol{Z}_1). \tag{A34}$$

Since $\boldsymbol{\Psi}$ is optimally obtained when solving IB problem (20), bottleneck constraint (20b) is thus satisfied and $I(\boldsymbol{x}; z_g|\boldsymbol{Z}_1) = R^{lb1}$. Then, from (A33) and (A34), we have

$$\begin{aligned}
I(\boldsymbol{y}; z_2|\boldsymbol{Z}_1) &\leq C - R(D), \\
I(\boldsymbol{x}; z_2|\boldsymbol{Z}_1) &\geq R^{lb1}.
\end{aligned} \tag{A35}$$

This completes the proof.

**Appendix E. Proof of Lemma 3**

When $M \to +\infty$, as stated in Appendix B, $\lambda - M \to 0$ almost surely. Then,

$$\int_{\frac{v}{\gamma}}^{\infty} \left( \log \frac{\gamma\lambda}{v} \right) f_\lambda(\lambda) d\lambda = \frac{C - R(D)}{T}$$

$$\to \log \frac{\gamma M}{v}, \tag{A36}$$

based on which we get

$$v - \gamma M 2^{-\frac{C-R(D)}{T}} \to 0. \tag{A37}$$

From (21), it is known that, as $M \to +\infty$,

$$R^{lb1} \to T \left[ \log(1 + \gamma M) - \log \left( 1 + \gamma M 2^{-\frac{C-R(D)}{T}} \right) \right]. \tag{A38}$$

It can be readily proven that $0 \le T \left[ \log(1 + \gamma M) - \log \left( 1 + \gamma M 2^{-\frac{C-R(D)}{T}} \right) \right] \le C$.

When $\rho \to +\infty$, $\sigma^2 \to 0$. Let $\gamma = \frac{1-D}{KD}$. $R^{lb1}$ thus tends to a constant and can be obtained from (21).

When $C \to +\infty$, it is possible for the relay to transmit $h$ almost perfectly to the destination node, i.e., $D \to 0$. Hence, $\gamma = \frac{1-D}{KD+\sigma^2} \to \rho$. In addition, it can be found from (22) that $v \to 0$. Then, from (21),

$$R^{lb1} \to T \int_0^{\infty} \log(1 + \rho\lambda) f_\lambda(\lambda) d\lambda$$

$$= I(x; y, H). \tag{A39}$$

Lemma 3 is thus proven.

**Appendix F. Proof of Theorem 3**

Since $\hat{n}_g \sim \mathcal{CN}(0, A_1')$ and $\lceil a_k \rceil_\mathcal{B}$ has $J$ possible values, i.e., $b_1, \cdots, b_J$, the channel in (32) can be divided into $KJ$ independent scalar Gaussian sub-channels with noise power $\lceil a_k \rceil_\mathcal{B} = b_j$ for each sub-channel. For the sub-channel with noise power $\lceil a_k \rceil_\mathcal{B} = b_j$, let $c_{k,j}$ denote the allocation of the bottleneck constraint $C$ and $R_{k,j}$ denote the corresponding rate. According to (A2), we have

$$R_{k,j} = \log(1 + \rho_j) - \log(1 + \rho_j 2^{-c_{k,j}}), \tag{A40}$$

where $\rho_j = \frac{1}{b_j}$. Since $b_J = +\infty$, we let $R_{k,J} = 0$ and $c_{k,J} = 0$. Note that, based on ([21], (16)), the representation of $\hat{x}_g$, i.e., $\hat{z}_g$, can be constructed by adding independent fading and Gaussian noise to each element of $\hat{x}_g$ in (32). Denote

$$P_{k,j} = \Pr\{ \lceil a_k \rceil_\mathcal{B} = b_j \}. \tag{A41}$$

Then, the optimal $I(x; \hat{z}_g | A_1')$ is equal to the objective function of the following problem:

$$\max_{\{c_{k,j}\}} \quad \sum_{k=1}^{K} \sum_{j=1}^{J-1} P_{k,j} R_{k,j} \tag{A42a}$$

$$\text{s.t.} \quad \sum_{k=1}^{K} \sum_{j=1}^{J-1} P_{k,j} c_{k,j} \le C - \sum_{k=1}^{K} H_k, \tag{A42b}$$

where $H_k = -\sum_{j=1}^{J} P_{k,j} \log P_{k,j}$.

Since $K \leq M$, as stated in Appendix A, $\boldsymbol{H}^H\boldsymbol{H} \sim \mathcal{CW}_K(M, \boldsymbol{I}_K)$. Matrix $(\boldsymbol{H}^H\boldsymbol{H})^{-1}$ thus follows complex inverse Wishart distribution and its diagonal elements are identically inverse chi squared distributed with $M - K + 1$ degrees of freedom [35]. Let $\eta$ denote one of the diagonal element of $(\boldsymbol{H}^H\boldsymbol{H})^{-1}$. The pdf of $\eta$ is thus given by

$$f_\eta(\eta) = \frac{2^{-(M-K+1)/2}}{\Gamma\left(\frac{M-K+1}{2}\right)} \eta^{-(M-K+1)/2-1} e^{-1/(2\eta)}. \tag{A43}$$

Since $\boldsymbol{A} = \sigma^2(\boldsymbol{H}^H\boldsymbol{H})^{-1}$, the diagonal entries of $\boldsymbol{A}$, i.e., $a_k$, $\forall k \in \mathcal{K}$, are marginally identically distributed. Let $a$ denote a new variable with the same distribution as $a_k$. $a$ thus follows the same distribution as $\sigma^2\eta$, and its pdf is given by

$$
\begin{aligned}
f_a(a) &= \frac{1}{\sigma^2} f_\eta\left(\frac{a}{\sigma^2}\right) \\
&= \frac{(2/\sigma^2)^{-(M-K+1)/2}}{\Gamma\left(\frac{M-K+1}{2}\right)} a^{-(M-K+1)/2-1} e^{-\sigma^2/(2a)}.
\end{aligned} \tag{A44}
$$

In addition, $P_{k,j}$, $R_{k,j}$, and $c_{k,j}$ can be simplified to $P_j$, $R_j$, and $c_j$ by dropping subscript "$k$". Using (A44), $P_j$ can be calculated as follows:

$$
\begin{aligned}
P_j &= \Pr\left\{ \lceil a \rceil_{\mathcal{B}} = b_j \right\} \\
&= \Pr\left\{ b_{j-1} < a \leq b_j \right\} \\
&= \int_{b_{j-1}}^{b_j} f_a(a)\,da.
\end{aligned} \tag{A45}
$$

Problem (A42) thus becomes

$$\max_{\{c_j\}} \quad \sum_{j=1}^{J-1} KP_jR_j \tag{A46a}$$

$$\text{s.t.} \quad \sum_{j=1}^{J-1} KP_jc_j \leq C - KH_0, \tag{A46b}$$

where

$$R_j = \log(1 + \rho_j) - \log(1 + \rho_j 2^{-c_j}),$$

$$H_0 = -\sum_{j=1}^{J} P_j \log P_j. \tag{A47}$$

Analogous to problem (A11), (A46) can be optimally solved by the water-filling method. The optimal $I(\boldsymbol{x}; \hat{\boldsymbol{z}}_g | \boldsymbol{A}_1')$ is given by

$$R^{lb2} = \sum_{j=1}^{J-1} KP_j\left[\log(1 + \rho_j) - \log(1 + \rho_j 2^{-c_j})\right]. \tag{A48}$$

where $c_j = \left[\log\frac{\rho_j}{\nu}\right]^+$ and $\nu$ is chosen such that the bottleneck constraint

$$\sum_{j=1}^{J-1} KP_jc_j = C - KH_0, \tag{A49}$$

is met. Theorem 3 is then proven.

**Appendix G. Proof of Lemma 4**

Since $\boldsymbol{\Phi}$ is a diagonal matrix with positive and real diagonal entries, it is invertible. Denote

$$\begin{aligned}
\boldsymbol{z}' &= \boldsymbol{\Phi}^{-1}\boldsymbol{z} \\
&= \boldsymbol{x} + \hat{\boldsymbol{n}} + \boldsymbol{\Phi}^{-1}\hat{\boldsymbol{n}}'_g, \\
\hat{\boldsymbol{z}}'_g &= \boldsymbol{\Phi}^{-1}\hat{\boldsymbol{z}}_g \\
&= \boldsymbol{x} + \hat{\boldsymbol{n}}_g + \boldsymbol{\Phi}^{-1}\hat{\boldsymbol{n}}'_g.
\end{aligned} \tag{A50}$$

For a given $\boldsymbol{A}'_1$, each element in $\hat{\boldsymbol{n}}$ is Gaussian distributed with zero mean and variance $\lceil a_k \rceil_{\mathcal{B}}$. However, $\hat{\boldsymbol{n}}$ is not a Gaussian vector since $\boldsymbol{H}$ is unknown. Hence, $\boldsymbol{z}'$ is not a Gaussian vector. For $\hat{\boldsymbol{z}}'_g$, from (32) and (41), it is known that $\hat{\boldsymbol{z}}'_g \sim \mathcal{CN}(\boldsymbol{0}, \boldsymbol{I}_K + \boldsymbol{A}'_1 + \boldsymbol{\Phi}^{-2})$.

We first prove inequation (43).

$$\begin{aligned}
I(\hat{\boldsymbol{x}}; \boldsymbol{z}|\boldsymbol{A}'_1) &= I(\hat{\boldsymbol{x}}; \boldsymbol{z}'|\boldsymbol{A}'_1) \\
&= h(\boldsymbol{z}'|\boldsymbol{A}'_1) - h(\boldsymbol{z}'|\hat{\boldsymbol{x}}, \boldsymbol{A}'_1) \\
&\overset{(a)}{\leq} \mathbb{E}\left[\log\det\left(\boldsymbol{I}_K + \mathbb{E}\left[\hat{\boldsymbol{n}}\hat{\boldsymbol{n}}^H\right] + \boldsymbol{\Phi}^{-2}\right) - \log\det\left(\boldsymbol{\Phi}^{-2}\right)\right] \\
&\overset{(b)}{\leq} \mathbb{E}\left[\log\det\left(\boldsymbol{I}_K + \boldsymbol{A}'_1 + \boldsymbol{\Phi}^{-2}\right) - \log\det\left(\boldsymbol{\Phi}^{-2}\right)\right] \\
&= I(\hat{\boldsymbol{x}}_g; \hat{\boldsymbol{z}}'_g|\boldsymbol{A}'_1) \\
&= I(\hat{\boldsymbol{x}}_g; \hat{\boldsymbol{z}}_g|\boldsymbol{A}'_1),
\end{aligned} \tag{A51}$$

where $(a)$ holds since Gaussian distribution maximizes the entropy over all distributions with the same variance and $(b)$ follows by using Hadamard's inequality.

Denote $\boldsymbol{x} = (x_1, \cdots, x_K)^T$, $\boldsymbol{z}' = (z'_1, \cdots, z'_K)^T$, $\hat{\boldsymbol{z}}'_g = (\hat{z}'_{g,1}, \cdots, \hat{z}'_{g,K})^T$, and $\boldsymbol{\Phi} = \text{diag}\{\varphi_1, \cdots, \varphi_K\}$. Then, we prove inequation (44). Using the chain rule of mutual information,

$$\begin{aligned}
I(\boldsymbol{x}; \boldsymbol{z}|\boldsymbol{A}'_1) &= I(\boldsymbol{x}; \boldsymbol{z}'|\boldsymbol{A}'_1) \\
&= \sum_{k=1}^{K} I(x_k; z'_k|\boldsymbol{A}'_1) + Q \\
&\geq \sum_{k=1}^{K} I(x_k; z'_k|\boldsymbol{A}'_1) \\
&\overset{(a)}{=} \sum_{k=1}^{K} I(x_k; \hat{z}'_{g,k}|\boldsymbol{A}'_1) \\
&\overset{(b)}{=} I(\boldsymbol{x}; \hat{\boldsymbol{z}}'_g|\boldsymbol{A}'_1) \\
&= I(\boldsymbol{x}; \hat{\boldsymbol{z}}_g|\boldsymbol{A}'_1),
\end{aligned} \tag{A52}$$

where $Q$ is a nonnegative constant; $(a)$ holds since for a given $\boldsymbol{A}'_1$, both $z'_k$ and $\hat{z}'_{g,k}$ follow $\mathcal{CN}\left(0, 1 + \lceil a_k \rceil_{\mathcal{B}} + \varphi_k^{-2}\right)$; and $(b)$ follows since the elements in $\boldsymbol{x}$ and $\hat{\boldsymbol{z}}'_g$ are independent.

Since $\boldsymbol{\Phi}$ is optimally obtained when solving IB problem (38), bottleneck constraint (38b) is thus satisfied and $I(\boldsymbol{x}; \hat{\boldsymbol{z}}_g|\boldsymbol{A}'_1) = R^{lb2}$. Then, from (A51) and (A52), we have

$$\begin{aligned}
I(\hat{\boldsymbol{x}}; \boldsymbol{z}|\boldsymbol{A}'_1) &\leq C - KH_0, \\
I(\boldsymbol{x}; \boldsymbol{z}|\boldsymbol{A}'_1) &\geq R^{lb2}.
\end{aligned} \tag{A53}$$

This completes the proof.

**Appendix H. Proof of Lemma 5**

As stated in Appendix B, when $M \to +\infty$, $\boldsymbol{H}^H\boldsymbol{H} - M\boldsymbol{I}_K \to \boldsymbol{0}$ almost surely. Hence,

$A - \frac{\sigma^2}{M} I_K \to 0$. Let $J = 2$, $b_1 = \frac{\sigma^2}{M} + \epsilon$, and $b_2 = +\infty$, where $\epsilon$ is a sufficiently small positive real number. Since $A - \frac{\sigma^2}{M} I_K \to 0$, we have $P_1 \to 1$ and $H_0 \to 0$. Then, from (39) and (40),

$$c_1 \to \frac{C}{K},$$

$$R^{lb2} \to K \left[ \log \left( 1 + \frac{M}{\sigma^2} \right) - \log \left( 1 + \frac{M}{\sigma^2} 2^{-\frac{C}{K}} \right) \right]$$

$$\to C. \tag{A54}$$

When $\rho \to +\infty$, $\sigma^2 \to 0$ and $A \to 0$. By setting $J = 2$ and $b_1$ small enough, it can be proven as above that $R^{lb2} \to C$.

When $C \to +\infty$, we could choose quantization points $\mathcal{B} = \{b_1, \cdots, b_J\}$ with sufficiently large $J$ such that the diagonal entries of $A_1$, which are continuously valued, can be represented precisely using the discretely valued points in $\mathcal{B}$, and the representation indexes of all diagonal entries can be transmitted to the destination node since $C$ is large enough. On the other hand, as shown in (41), a representation of $\hat{x}_g$ is

$$\hat{z}_g = \Phi \hat{x}_g + \hat{n}'_g, \tag{A55}$$

where $\Phi$ is a diagonal matrix with positive and real diagonal entries, and $\hat{n}'_g \sim \mathcal{CN}(0, I_K)$. As $C \to +\infty$, according to ([21], (17) and (20)), the diagonal entries of $\Phi$

$$\varphi_k = \sqrt{\frac{\frac{1}{\lceil a_k \rceil_\mathcal{B}} + 2^C}{1 + \lceil a_k \rceil_\mathcal{B}} - \frac{1}{\lceil a_k \rceil_\mathcal{B}}}$$

$$\to \sqrt{\frac{2^C}{1 + \lceil a_k \rceil_\mathcal{B}}}, \quad \forall k \in \mathcal{K}. \tag{A56}$$

Since $\Phi$ is a diagonal matrix with positive and real diagonal entries, as in (A50), we can get

$$\hat{z}'_g = \Phi^{-1} \hat{z}_g$$

$$= \hat{x}_g + \Phi^{-1} \hat{n}'_g. \tag{A57}$$

From (A56), it is known that the elements in noise vector $\Phi^{-1} \hat{n}'_g$ have zero mean and very small (approaches 0) power when $C \to +\infty$. Hence, $(x, \hat{z}'_g) \to (x, \hat{x}_g)$ in distribution. Then, based on [36], we have

$$I(x; \hat{x}_g | A'_1) \leq \liminf_{C \to +\infty} I(x; \hat{z}'_g | A'_1). \tag{A58}$$

In addition, since Gaussian noise vector $\hat{n}_g$ (defined in (32)) is independent of $x$ and $\Phi^{-1} \hat{n}'_g$ in (A57) is independent of both $x$ and $\hat{n}_g$, $x \to \hat{x}_g \to \hat{z}'_g$ forms a Markov Chain. Then, according to data-processing inequality, we have

$$I(x; \hat{z}'_g | A'_1) \leq I(x; \hat{x}_g | A'_1). \tag{A59}$$

Combining (A59) and (A58), we have

$$I(x; \hat{x}_g | A'_1) \leq \liminf_{C \to +\infty} I(x; \hat{z}'_g | A'_1) \leq I(x; \hat{x}_g | A'_1), \tag{A60}$$

showing that the limit $\liminf\limits_{C\to+\infty} I(x;\hat{z}'_g|A'_1)$ exists and that it is equal to $I(x;\hat{x}_g|A'_1)$. Then, when $C\to+\infty$,

$$
\begin{aligned}
R^{lb2} &= I(x;\hat{z}_g|A'_1) \\
&= I(x;\hat{z}'_g|A'_1) \\
&\to I(x;\hat{x}_g|A'_1) \\
&= \mathbb{E}\big[\log\det(I_K + A'_1) - \log\det(A'_1)\big] \\
&\to \mathbb{E}[\log\det(I_K + A_1) - \log\det(A_1)],
\end{aligned}
\tag{A61}
$$

On the other hand, the capacity of Channel 1 is given by

$$
\begin{aligned}
I(x;y,H) &= I(x;y|H) \\
&= \mathbb{E}\left[\log\det\left(HH^H + \sigma^2 I_M\right) - \log\det\left(\sigma^2 I_M\right)\right] \\
&= \mathbb{E}\left[\log\det\left(H^H H + \sigma^2 I_K\right) - \log\det\left(\sigma^2 I_K\right)\right] \\
&= \mathbb{E}[\log\det(I_K + A) - \log\det(A)].
\end{aligned}
\tag{A62}
$$

To prove that (A61) is upper bounded by (A62), we first give and prove the following lemma.

**Lemma A1.** *For any K-dimensional positive definite matrix $N$, let $N_1 = N \odot I_K$, i.e., $N_1$ consist of the diagonal elements of $N$. Then,*

$$
\log\det(I_K + N) - \log\det(N) \geq \log\det(I_K + N_1) - \log\det(N_1).
\tag{A63}
$$

**Proof.** Obviously, (A63) is equivalent to

$$
\log\det(N_1) - \log\det(N) \geq \log\det(I_K + N_1) - \log\det(I_K + N).
\tag{A64}
$$

To prove (A64), we introduce an auxiliary function $g_1(x) = \log\det(xI_K + N_1) - \log\det(xI_K + N)$ and show that $g_1(x)$ decreases monotonically with respect to $x$ when $x \geq 0$. By taking the first-order derivative to $g_1(x)$, we have

$$
g'_1(x) = \text{tr}\left[(xI_K + N_1)^{-1}\right] - \text{tr}\left[(xI_K + N)^{-1}\right].
\tag{A65}
$$

To prove $g'_1(x) \leq 0$, we show in the following that, for any positive definite matrix $O$, we always have

$$
\text{tr}\left(O_1^{-1}\right) \leq \text{tr}\left(O^{-1}\right),
\tag{A66}
$$

where $O_1$ consists of the diagonal elements of $O$, i.e., $O_1 = O \odot I_K$. Denote the diagonal entries of $O$ (or $O_1$) by $o = (o_1, \cdots, o_K)^T$ and the eigenvalues of $O$ by $\theta = (\theta_1, \cdots, \theta_K)^T$. Since $O$ is a positive definite matrix, the entries of $o$ and $\theta$ are real and positive. In addition, according to the Schur–Horn theorem, $o$ is majorized by $\theta$, i.e.,

$$
o \prec \theta.
\tag{A67}
$$

Define a real vector $u = (u_1, \cdots, u_K)^T$ with $u_k > 0$, $\forall k \in \mathcal{K}$, and function $g_2(u) = \sum_{k=1}^{K} \frac{1}{u_k}$. It is obvious that $g_2(u)$ is convex and symmetric. Hence, $g_2(u)$ is a Schur-convex function. Therefore,

$$
g_2(o) \leq g_2(\theta).
\tag{A68}
$$

Using (A68), we have

$$\text{tr}\left(\boldsymbol{O}_1^{-1}\right) = \sum_{k=1}^{K} \frac{1}{o_k}$$
$$= g_2(\boldsymbol{o})$$
$$\leq g_2(\boldsymbol{\theta})$$
$$= \sum_{k=1}^{K} \frac{1}{\theta_k}$$
$$= \text{tr}\left(\boldsymbol{O}^{-1}\right), \tag{A69}$$

based on which we get $g_1'(x) \leq 0$ and (A63) can then be proven. $\quad\square$

Then, from (A61), (A62), and Lemma A1, it is known that, when $C \to +\infty$,

$$R^{lb2} \to \mathbb{E}[\log\det(\boldsymbol{I}_K + \boldsymbol{A}_1) - \log\det(\boldsymbol{A}_1)]$$
$$= K\mathbb{E}\left[\log\left(1 + \frac{1}{a}\right)\right]$$
$$\leq I(\boldsymbol{x}; \boldsymbol{y}, \boldsymbol{H}), \tag{A70}$$

where the expectation can be calculated by using the pdf of $a$ in (A44). Lemma 5 is thus proven.

**Appendix I. Proof of Remark 2**

In this appendix, we show that, when $K = M$ and $\lambda_{\text{th}} = 0$, $\mathbb{E}\left[\frac{1}{\lambda}\right]$ does not exist.

When $K = M$, $f_\lambda(\lambda)$ is given in (A17). From (A18), it is known that, for any $0 \leq i \leq K - 1$, $L_i^0$ can always be expressed as follows:

$$L_i^0 = \frac{e^\lambda}{i!} \frac{d^i}{d\lambda^i}\left(e^{-\lambda}\lambda^i\right)$$
$$= \sum_{j=1}^{i} \varsigma_{i,j}\lambda^j + 1, \tag{A71}$$

where $\varsigma_{i,j}$ is a constant. Accordingly, from (A17),

$$f_\lambda(\lambda) = \frac{1}{K} \sum_{i=0}^{K-1} \left[L_i^0(\lambda)\right]^2 e^{-\lambda}$$
$$= \frac{e^{-\lambda}}{K} \sum_{j=1}^{2(K-1)} \tau_j\lambda^j + e^{-\lambda}, \tag{A72}$$

where $\tau_j$ is a constant. Let $\epsilon$ denote a sufficiently small positive real number. Then, when $\lambda_{\text{th}} = 0$,

$$\mathbb{E}\left[\frac{1}{\lambda}\right] = \int_0^\infty \frac{1}{\lambda} f_\lambda(\lambda) d\lambda$$
$$= \int_0^\infty \frac{e^{-\lambda}}{K} \sum_{j=1}^{2(K-1)} \tau_j\lambda^{j-1} d\lambda + \int_0^\infty \frac{1}{\lambda} e^{-\lambda} d\lambda$$
$$= \frac{1}{K} \sum_{j=1}^{2(K-1)} \tau_j(j-1)! - \text{Ei}(-0), \tag{A73}$$

where we used $\int_0^\infty e^{-\lambda}\lambda^{j-1}d\lambda = (j-1)!$ and $\text{Ei}(\cdot)$ is the exponential integral. As is well-known, $lim_{x\to 0} - \text{Ei}(-x) = \infty$. Hence, the integral in (A73) diverges. $\mathbb{E}\left[\frac{1}{\lambda}\right]$ thus does

not exist.

**Appendix J. Proof of Lemma 7**

As stated in Appendix B, when $M \to +\infty$, $H^H H - M I_K \to 0$ almost surely. Hence,

$$
\frac{1}{\lambda} \to 0,
$$

$$
P_{\text{th}} = \Pr\{\lambda_{\min} \geq \lambda_{\text{th}}\}
$$

$$
\to 1,
$$

$$
H_{\text{th}} \to 0,
$$

$$
D \to \frac{1}{2^{\frac{C}{K}} - 1},
$$

$$
\frac{1}{\mathbb{E}[\lambda|\Delta]} \to 0,
$$

$$
\mathbb{E}\left[\frac{1}{\lambda}\Big|\Delta\right] \to 0. \tag{A74}
$$

Combining (A74) with (66)–(68), we have

$$
R^{lb3}, \check{R}^{lb3}, \hat{R}^{lb3} \to K \log\left(1 + \frac{1}{D}\right)
$$

$$
\to C. \tag{A75}
$$

When $\rho \to +\infty$, $\sigma^2 \to 0$. Hence,

$$
D \to \frac{1}{2^{\frac{C - H_{\text{th}}}{P_{\text{th}} K}} - 1},
$$

$$
R^{lb3}, \check{R}^{lb3}, \hat{R}^{lb3} \to P_{\text{th}} K \log\left(1 + \frac{1}{D}\right)
$$

$$
\to C - H_{\text{th}}. \tag{A76}
$$

When $C \to +\infty$, it can be found from (62) that $D \to 0$. Then, from (66)–(68), it is known that $R^{lb3}$, $\check{R}^{lb3}$, and $\hat{R}^{lb3}$ all approach constants, which can be, respectively, obtained by setting $D = 0$ in (66)–(68). Lemma 7 is thus proven.

**Appendix K. Proof of Theorem 5**

As stated in Appendix A, $U \Lambda U^H$ is the eigendecomposition of $H H^H$ and $\lambda_t, \forall t \in \mathcal{T}$ are unordered positive eigenvalues of $H H^H$. To derive $R^{lb4}$, we further denote the singular value decomposition of $H$ by $U L V^H$, where $V \in \mathbb{C}^{K \times K}$ is a unitary matrix and $L \in \mathbb{R}^{M \times K}$ is a rectangular diagonal matrix. In fact, the diagonal entries of $L$ are the nonnegative square roots of the positive eigenvalues of $H H^H$. Then, from (73), we have

$$
\begin{aligned}
\boldsymbol{F}^H \boldsymbol{H} =& \boldsymbol{H}^H \left( \boldsymbol{H}\boldsymbol{H}^H + \sigma^2 \boldsymbol{I}_M \right)^{-1} \boldsymbol{H}, \\
=& \boldsymbol{V}\boldsymbol{L}^H \left( \boldsymbol{\Lambda} + \sigma^2 \boldsymbol{I}_M \right)^{-1} \boldsymbol{L}\boldsymbol{V}^H, \\
=& \boldsymbol{V}\mathrm{diag}\left\{ \frac{\lambda_1}{\lambda_1 + \sigma^2}, \cdots, \frac{\lambda_T}{\lambda_T + \sigma^2}, \boldsymbol{0}_{K-T}^H \right\} \boldsymbol{V}^H, \\
\boldsymbol{F}^H \boldsymbol{H}\boldsymbol{H}^H \boldsymbol{F} =& \boldsymbol{V}\boldsymbol{L}^H \left( \boldsymbol{\Lambda} + \sigma^2 \boldsymbol{I}_M \right)^{-1} \boldsymbol{\Lambda} \left( \boldsymbol{\Lambda} + \sigma^2 \boldsymbol{I}_M \right)^{-1} \boldsymbol{L}\boldsymbol{V}^H, \\
=& \boldsymbol{V}\mathrm{diag}\left\{ \frac{\lambda_1^2}{(\lambda_1 + \sigma^2)^2}, \cdots, \frac{\lambda_T^2}{(\lambda_T + \sigma^2)^2}, \boldsymbol{0}_{K-T}^H \right\} \boldsymbol{V}^H, \\
\boldsymbol{F}^H \boldsymbol{F} =& \boldsymbol{V}\boldsymbol{L}^H \left( \boldsymbol{\Lambda} + \sigma^2 \boldsymbol{I}_M \right)^{-2} \boldsymbol{L}\boldsymbol{V}^H, \\
=& \boldsymbol{V}\mathrm{diag}\left\{ \frac{\lambda_1}{(\lambda_1 + \sigma^2)^2}, \cdots, \frac{\lambda_T}{(\lambda_T + \sigma^2)^2}, \boldsymbol{0}_{K-T}^H \right\} \boldsymbol{V}^H,
\end{aligned}
\tag{A77}
$$

where $\boldsymbol{0}_{K-T}$ is a $(K-T)$-dimensional all "0" column vector. Based on (A77),

$$
\begin{aligned}
& \boldsymbol{F}^H \boldsymbol{H}\boldsymbol{H}^H \boldsymbol{F} + \sigma^2 \boldsymbol{F}^H \boldsymbol{F} + D\boldsymbol{I}_K \\
=& \boldsymbol{V}\mathrm{diag}\left\{ \frac{\lambda_1}{\lambda_1 + \sigma^2} + D, \cdots, \frac{\lambda_T}{\lambda_T + \sigma^2} + D, D \times \boldsymbol{1}_{K-T}^H \right\} \boldsymbol{V}^H,
\end{aligned}
\tag{A78}
$$

where $\boldsymbol{1}_{K-T}$ is a $(K-T)$-dimensional all "1" column vector. Since $\boldsymbol{\Lambda}$ is independent of $\boldsymbol{U}$, $\boldsymbol{L}$ is independent of $\boldsymbol{U}$ as well as $\boldsymbol{V}$, and $\lambda_t, \forall t \in \mathcal{T}$ is unordered, we have

$$
\begin{aligned}
& \mathbb{E}\left[ \log \det \left( \boldsymbol{F}^H \boldsymbol{H}\boldsymbol{H}^H \boldsymbol{F} + \sigma^2 \boldsymbol{F}^H \boldsymbol{F} + D\boldsymbol{I}_K \right) \right] \\
=& T\mathbb{E}\left[ \log \left( \frac{\lambda}{\lambda + \sigma^2} + D \right) \right] + (K - T) \log D.
\end{aligned}
\tag{A79}
$$

Then, we calculate $\boldsymbol{G}$ in (82). For this purpose, we have to calculate $\mathbb{E}\left[ \boldsymbol{F}^H \boldsymbol{H} \right]$, $\mathbb{E}\left[ \boldsymbol{F}^H \boldsymbol{H}\boldsymbol{H}^H \boldsymbol{F} \right]$, and $\mathbb{E}\left[ \boldsymbol{F}^H \boldsymbol{F} \right]$. To obtain these expectations, we consider two different cases, i.e., the case with $K \le M$ and the case with $K > M$. When $K \le M$, from (A77), we have

$$
\begin{aligned}
\mathbb{E}\left[ \boldsymbol{F}^H \boldsymbol{H} \right] &= \mathbb{E}\left[ \frac{\lambda}{\lambda + \sigma^2} \right] \boldsymbol{I}_K, \\
\mathbb{E}\left[ \boldsymbol{F}^H \boldsymbol{H}\boldsymbol{H}^H \boldsymbol{F} \right] &= \mathbb{E}\left[ \frac{\lambda^2}{(\lambda + \sigma^2)^2} \right] \boldsymbol{I}_K, \\
\mathbb{E}\left[ \boldsymbol{F}^H \boldsymbol{F} \right] &= \mathbb{E}\left[ \frac{\lambda}{(\lambda + \sigma^2)^2} \right] \boldsymbol{I}_K.
\end{aligned}
\tag{A80}
$$

When $K > M$, denote $\boldsymbol{V} = (\boldsymbol{v}_1, \cdots, \boldsymbol{v}_K)$. Then, from (A77),

$$
\begin{aligned}
\boldsymbol{F}^H \boldsymbol{H} &= \boldsymbol{V}\mathrm{diag}\left\{ \frac{\lambda_1}{\lambda_1 + \sigma^2}, \cdots, \frac{\lambda_M}{\lambda_M + \sigma^2}, \boldsymbol{0}_{K-T}^H \right\} \boldsymbol{V}^H \\
&= \left( \frac{\lambda_1}{\lambda_1 + \sigma^2} \boldsymbol{v}_1, \cdots, \frac{\lambda_M}{\lambda_M + \sigma^2} \boldsymbol{v}_M, \boldsymbol{0}_K^H, \cdots, \boldsymbol{0}_K^H \right)
\begin{bmatrix} \boldsymbol{v}_1^H \\ \vdots \\ \boldsymbol{v}_K^H \end{bmatrix} \\
&= \sum_{m=1}^M \frac{\lambda_m}{\lambda_m + \sigma^2} \boldsymbol{v}_m \boldsymbol{v}_m^H.
\end{aligned}
\tag{A81}
$$

Since $v_m$ is the eigenvector of matrix $H^H H$ and is independent of unordered eigenvalue $\lambda_m$, we have

$$\mathbb{E}\Big[F^H H\Big] = \sum_{m=1}^{M} \mathbb{E}\left[\frac{\lambda_m}{\lambda_m + \sigma^2}\right] \frac{1}{K} I_K$$

$$= \frac{M}{K} \mathbb{E}\left[\frac{\lambda}{\lambda + \sigma^2}\right] I_K. \tag{A82}$$

Similarly, we also have

$$\mathbb{E}\Big[F^H H H^H F\Big] = \frac{M}{K} \mathbb{E}\left[\frac{\lambda^2}{(\lambda + \sigma^2)^2}\right] I_K,$$

$$\mathbb{E}\Big[F^H F\Big] = \frac{M}{K} \mathbb{E}\left[\frac{\lambda}{(\lambda + \sigma^2)^2}\right] I_K. \tag{A83}$$

Using (A80), (A82), (A83), and (82), $G$ can be calculated as

$$G = \mathbb{E}\Big[F^H H H^H F\Big] - \mathbb{E}\Big[F^H H\Big]\mathbb{E}\Big[H^H F\Big] + \sigma^2 \mathbb{E}\Big[F^H F\Big] + D I_K$$

$$= \left\{ \frac{T}{K} \mathbb{E}\left[\frac{\lambda}{\lambda + \sigma^2}\right] - \frac{T^2}{K^2}\left(\mathbb{E}\left[\frac{\lambda}{\lambda + \sigma^2}\right]\right)^2 + D \right\} I_K. \tag{A84}$$

Hence,

$$\log \det(G) = K \log \left\{ \frac{T}{K} \mathbb{E}\left[\frac{\lambda}{\lambda + \sigma^2}\right] - \frac{T^2}{K^2}\left(\mathbb{E}\left[\frac{\lambda}{\lambda + \sigma^2}\right]\right)^2 + D \right\}. \tag{A85}$$

Substituting (A79) and (A85) into (80) and (81), respectively, and using (79), we can get (83). We then calculate $D$ in (84). From (77), (A80), and (A83),

$$\mathbb{E}\Big[\bar{x}\bar{x}^H\Big] = \mathbb{E}\Big[F^H H H^H F + \sigma^2 F^H F\Big]$$

$$= \frac{T}{K} \mathbb{E}\left[\frac{\lambda}{\lambda + \sigma^2}\right] I_K. \tag{A86}$$

$I(\bar{x}_g; \bar{z}_g)$ in (78) can thus be calculated as follows:

$$I(\bar{x}_g; \bar{z}_g) = \log \det \left( I_K + \frac{\mathbb{E}\Big[\bar{x}\bar{x}^H\Big]}{D} \right)$$

$$= K \log \left( 1 + \frac{T}{DK} \mathbb{E}\left[\frac{\lambda}{\lambda + \sigma^2}\right] \right)$$

$$= C, \tag{A87}$$

based on which (84) can be obtained. Theorem 5 is then proven.

**Appendix L. Proof of Lemma 9**

When $M \to +\infty$, $T = K$. As stated in Appendix B, $H^H H - M I_K \to 0$ almost surely. Hence, $\lambda - M \to 0$. From (A87),

$$I(\bar{x}_g; \bar{z}_g) = K \log \left( 1 + \frac{1}{D} \mathbb{E}\left[\frac{\lambda}{\lambda + \sigma^2}\right] \right)$$

$$= C$$

$$\to K \log \left( 1 + \frac{1}{D} \right). \tag{A88}$$

Combining (83) and (A88), we have

$$R^{lb4} \rightarrow K\log(1+D) - K\log D$$
$$= K\log\left(1+\frac{1}{D}\right)$$
$$\rightarrow C. \tag{A89}$$

When $K \leq M$ and $\rho \rightarrow +\infty$, $T = K$ and $\sigma^2 \rightarrow 0$. Using (A87) and (83), we can also get (A88) and (A89).

When $K \leq M$ and $C \rightarrow +\infty$, it can be found from (84) that $D \rightarrow 0$. Then, using (83), we can get (85). This finishes the proof.

**Appendix M. Proof of Lemma 10**

As shown in Lemmas 3 and 5, when $C \rightarrow +\infty$, $R^{lb1}$ approaches the capacity of Channel 1 while $R^{lb2}$ is upper bounded by the capacity of Channel 1. Hence,

$$R^{lb1} \geq R^{lb2}. \tag{A90}$$

Moreover, as shown in (52), we quantize $\tilde{x}$ by adding Gaussian noise vector $q \sim \mathcal{CN}(\mathbf{0}, D\mathbf{I}_K)$ when event $\Delta$ happens and obtain its representation $z$. When $C \rightarrow +\infty$, it is known from (62) that $D \rightarrow 0$. Hence, $(x, z) \rightarrow (x, \tilde{x})$ in distribution, and it can be proven similarly to (A61) that

$$R^{lb3} \leq P_{\text{th}} I(x; z|\Delta)$$
$$\rightarrow P_{\text{th}} I(x; \tilde{x}|\Delta). \tag{A91}$$

Using (A39) and (A91), we have

$$R^{lb1} \rightarrow I(x; y, H)$$
$$= h(x) - h(x|y, H)$$
$$= h(x) - h(x|y, H, \tilde{x})$$
$$\geq h(x) - h(x|\tilde{x})$$
$$= I(x; \tilde{x})$$
$$\geq P_{\text{th}} I(x; \tilde{x}|\Delta)$$
$$\rightarrow P_{\text{th}} I(x; z|\Delta)$$
$$\geq R^{lb3}. \tag{A92}$$

Analogously, from (76) and (84), it is known that $(x, z) \rightarrow (x, \bar{x})$ in distribution when $C \rightarrow +\infty$. Hence,

$$R^{lb1} \rightarrow I(x; y, H)$$
$$= h(x) - h(x|y, H)$$
$$= h(x) - h(x|y, H, \bar{x})$$
$$\geq h(x) - h(x|\bar{x})$$
$$\geq I(x; \bar{x})$$
$$\rightarrow I(x; z)$$
$$\geq R^{lb4}, \tag{A93}$$

where $\bar{x}$ is the MMSE estimate of $x$ at the relay, i.e., (74). This completes the proof.

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
