# Peer review of "Information Bottleneck for a Rayleigh Fading MIMO Channel with an Oblivious Relay"

_information, doi:10.3390/info12040155_

Round 1

Reviewer 1 Report

This paper discusses the information bottleneck problem at a relay for independent Rayleigh fading channels with multiple antennas. The paper is very well written and the results are scientifically sound, as well as correct (as far as I could tell from the manuscript). I recommend publication.

I have very minor comments:

  • Perhaps this is my major (minor) comment: the figures are very hard to read on black and white printed paper. Not only because of the colors (actually, each curve has its own marker, so it is OK), but because of the size of the figure and the number of curves reported. I would strongly suggest to a) make all text in the figures bigger (same font size as the text), b) make the figures bigger in size (this is key -- to read the curves I had to open the pdf and zoom on the figures, which is largely undesirable for a paper, unless one wants to zoom into a particular effect), and c) use either the Latex interpreted of Matlab or use tikz (matlab2tikz) for an improved formatting.
  • line 39 can not --> cannot
  • line 62: in [26] we studied.... this was done by a subset of the authors, so I suggest a rephrase
  • define \rho (the snr) in the model, not buried in Theorem 1
  • H_{sum_indi} should be redefined somehow as this is ugly notation. Please find something more appropriate. Even H_sum works much better
  • What happens for small dimensions including C (on the order of 4-6)? Do the same relative comparisons  hold? Most results reported are for large dimensions and large C (which is perhaps the natural operating point of these systems), but it would be of interest (especially academic) to understand the small dimension case, as it might highlight additional limitations of some schemes that are not visible in the high dimensional case, but which, if overcome, could improve the performance (even for high dimensions).

I have other comments unrelated to the technical contribution/presentation of the paper and do not affect my appreciation of the work.

  • The model feels a bit contrived. This might be due to the fact that I do not actively work on IB problems, but it does not feel natural... Not only the IB problem feels unnatural, but the assumptions are odd: independent fading, perfect CSI, sending compressed (noisy) CSI to the destination... Taking into account the Doppler spread of the fading process and some sort of channel estimation (via pilots, or noncoherent transmission altogether) at the relay might make more sense, as it is difficult to assess how realistic are the conclusions extracted from the relative comparisons between schemes and the upper bound.
  • How good/realistic is the upper bound? It is unclear (at least from the study) whether the upper bound can be improved -- it would seem to me that it can, but of course, with additional assumptions. Perhaps letting the destination know the channel up to the same noise error that the compression at the relay (but different realization of course). It is hard to tell for me whether this makes sense, but at the same time, it is difficult to understand how good the results are.

Author Response

Please see the pdf file attached

Author Response

Please see the pdf file attached

Round 2

Reviewer 2 Report

The authors have carefully revised the manuscript and addressed all of my comments. However, I still have a number of doubts:

1) Proof of Lemma 1 in Appendix B: I believe the last step in (107) ((R3) in the reviewer response) needs to be done a bit more carefully. Indeed, the authors divide the integral into the parts |\lambda-M| \leq \epsilon and |\lambda-M| > \epsilon. While it is true that the probability of the second part vanishes as M\to\infty, the integrand is the pdf times log(\rho\lambda/\nu), which is unbounded in \lambda.

2) Proof of Lemma 5: The authors first show that \hat{x}_g converges to \hat{z}'_g, and then they use that "mutual information is a mapping from joint distributions to a real value" to conclude that the corresponding mutual informations converge. I did not understand the "mapping from joint distributions to a real value" argumentation. I agree that mutual information is a function of a joint distribution, say I(P). However, even if P' converges to P'' in distribution, this does not necessarily imply that I(P') converges to I(P''), because this is only true if the mapping is continuous around P'', and mutual information is in general not continuous (it is lower semicontinuous).

3) Similarly, my doubts about the proof of Lemma 10 still remain.
